# No-Regret Learning with Unbounded Losses: The Case of Logarithmic Pooling

**Eric Neyman**
Columbia University
New York, NY 10027
eric.neyman@columbia.edu

**Tim Roughgarden**
Columbia University
New York, NY 10027
tim.roughgarden@gmail.com

## Abstract

For each of $T$ time steps, $m$ experts report probability distributions over $n$ outcomes; we wish to learn to aggregate these forecasts in a way that attains a no-regret guarantee. We focus on the fundamental and practical aggregation method known as *logarithmic pooling* — a weighted average of log odds — which is in a certain sense the optimal choice of pooling method if one is interested in minimizing log loss (as we take to be our loss function). We consider the problem of learning the best set of parameters (i.e. expert weights) in an online adversarial setting. We assume (by necessity) that the adversarial choices of outcomes and forecasts are consistent, in the sense that experts report calibrated forecasts. Imposing this constraint creates a (to our knowledge) novel semi-adversarial setting in which the adversary retains a large amount of flexibility. In this setting, we present an algorithm based on online mirror descent that learns expert weights in a way that attains $O(\sqrt{T} \log T)$ expected regret as compared with the best weights in hindsight.

## 1 Introduction

### 1.1 Logarithmic pooling

Suppose that $m$ experts report probability distributions $\mathbf{p}^1, \ldots, \mathbf{p}^m \in \Delta^n$ over $n$ disjoint, exhaustive outcomes. We are interested in aggregating these distributions into a single distribution $\mathbf{p}^*$, a task known as probabilistic *opinion pooling*. Perhaps the most straightforward way to do this is to take the arithmetic mean, also called the linear pool: $\mathbf{p}^* = \frac{1}{m} \sum_{i=1}^m \mathbf{p}^i$.

While commonly studied and frequently used, the linear pool is by no means the definitive opinion pooling method. The choice of pooling method ought to depend on context, and in particular on the loss function with respect to which forecasts are assessed. Neyman and Roughgarden (2023) showed a correspondence between proper loss functions and opinion pooling methods, which they termed *quasi-arithmetic (QA) pooling* with respect to a loss function. Specifically, the QA pool with respect to a loss is the forecast that guarantees the largest possible overperformance (as judged by the loss function) compared with the strategy of choosing a random expert to trust.[1]

In this work we will be using log loss function. The QA pooling technique with respect to the log loss is known as *logarithmic pooling*. Instead of averaging the experts' forecasts, logarithmic pooling averages the experts' log odds (i.e. logits). Put otherwise, the logarithmic pool is defined by

$$p_j^* = c \prod_{i=1}^m (p_j^i)^{1/m}$$

---

[1]More formally, the QA pool maximizes the minimum (over possible outcomes) improvement in the loss.

for all events $j \in [n]$, where $c$ is a normalizing constant to ensure that the probabilities add to $1$. While the linear pool is an arithmetic mean, the logarithmic pool behaves much more like a geometric mean. For instance, the logarithmic pool of $(0.001, 0.999)$ and $(0.5, 0.5)$ with equal weights is approximately $(0.03, 0.97)$.

The logarithmic pool has been studied extensively because of its naturalness (Genest, 1984; Genest and Zidek, 1986; Givens and Roback, 1999; Poole and Raftery, 2000; Kascha and Ravazzolo, 2008; Rufo et al., 2012; Allard et al., 2012). Logarithmic pooling can be interpreted as averaging experts' Bayesian evidence (Neyman and Roughgarden, 2023, §A). It is also the most natural pooling method that satisfies *external Bayesianality*, meaning roughly that it does not matter whether a Bayesian update is applied to each expert's forecast before aggregation, or if instead the Bayesian update is applied to the pooled forecast (Genest, 1984). Logarithmic pooling can also be characterized as the pooling method that minimizes the average KL divergence to the experts' reports (Abbas, 2009). Finally, empirical work has found logarithmic pooling performs very well on real-world data (Satopää et al., 2014; Sevilla, 2021).

## 1.2 Logarithmic pooling with weighted experts

Forecast aggregators often assign different weights to different experts, e.g. based on each expert's level of knowledge or track record (Tetlock and Gardner, 2015). There is a principled way to include weights $w_1, \dots, w_i$ (summing to $1$) in the logarithmic pool, namely:

$$p_j^*(\mathbf{w}) = c(\mathbf{w}) \prod_{i=1}^{m} (p_j^i)^{w_i},$$

where $c(\mathbf{w})$ is again a normalizing constant that now depends on the weights.[2] This more general notion continues to have all of the aforementioned natural properties.

The obvious question is: how does one know what these weights should be? Perhaps the most natural answer is to weight experts according to their past performance. Finding appropriate weights for experts is thus an online learning problem. This learning problem is the focus of our work.

## 1.3 Choosing the right benchmark

Our goal is to develop an algorithm for learning weights for logarithmic pooling in a way that achieves vanishing regret as judged by the log loss function (i.e. the loss function most closely associated with this pooling method). Within the field of online prediction with expert advice, this is a particularly challenging setting. In part, this is because the losses are potentially unbounded. However, that is not the whole story: finding weights for *linear* pooling, by contrast, is a well-studied problem that has been solved even in the context of log loss. On the other hand, because logarithmic pooling behaves more as a geometric than an arithmetic mean, if some expert assigns a very low probability to the eventual outcome (and the other experts do not) then the logarithmic pool will also assign a low probability, incurring a large loss. This makes the combination of logarithmic pooling with log loss particularly difficult.

We require that our algorithm not have access to the experts' forecasts when choosing weights: an algorithm that chooses weights in a way that depends on forecasts can output an essentially arbitrary function of the forecasts, and thus may do something other than learn optimal weights for logarithmic pooling. For example, suppose that $m = n = 2$ and an aggregator wishes to subvert our intentions and take an equally weighted *linear* pool of the experts' forecasts. Without knowing the experts' forecasts, this is impossible; on the other hand, if the aggregator knew that e.g. $\mathbf{p}_1 = (90\%, 10\%)$ and $\mathbf{p}_2 = (50\%, 50\%)$, they could assign weights for logarithmic pooling so as to produce the post-hoc desired result, i.e. $(70\%, 30\%)$. We wish to disallow this.

One might suggest the following setup: at each time step, the algorithm selects weights for each expert. Subsequently, an adversary chooses each expert's forecast and the outcome, after which the algorithm and each expert incur a log loss. Unfortunately — due to the unboundedness of log loss and the behavior of logarithmic pooling — vanishing regret guarantees in this setting are impossible.

---

[2]Were it not for the normalizing constant, the log loss incurred by the logarithmic pool $p_j^*(\mathbf{w})$ would be a linear function of the weights. However, the $c(\mathbf{w})$ term significantly complicates this picture.

**Example 1.1.** Consider the case of $m = n = 2$. Without loss of generality, suppose that the algorithm assigns Expert 1 a weight $w \geq 0.5$ in the first time step. The adversary chooses reports $(e^{-T}, 1 - e^{-T})$ for Expert 1 and $\left(\frac{1}{2}, \frac{1}{2}\right)$ for Expert 2, and for Outcome 1 to happen. The logarithmic pool of the forecasts turns out to be approximately $(e^{-wT}, 1 - e^{-wT})$, so the algorithm incurs a log loss of approximately $wT \geq 0.5T$, compared to $O(1)$ loss for Expert 2. On subsequent time steps, Expert 2 is perfect (assigns probability 1 to the correct outcome), so the algorithm cannot catch up.

What goes wrong in Example 1.1 is that the adversary has full control over experts' forecast *and* the realized outcome, and is not required to couple the two in any way. This unreasonable amount of adversarial power motivates assuming that the experts are *calibrated*: for example, if an expert assigns a 10% chance to an outcome, there really is a 10% chance of that outcome (conditional on the expert's information).

We propose the following setting: an adversary chooses a joint probability distribution over the experts' beliefs and the outcome — subject to the constraint that each expert is calibrated. The adversary retains full control over correlations between forecasts and outcomes, subject to the calibration property. Subsequently, nature randomly samples each expert's belief and the eventual outcome from the distribution. In this setting, we seek to prove upper bounds on the expected value of our algorithm's regret.

Why impose this constraint, instead of a different one? Our reasons are twofold: theoretical and empirical. From a theoretical standpoint, the assumption that experts are calibrated is natural because experts who form Bayesian rational beliefs based on evidence will be calibrated, regardless of how much or how little evidence they have. The assumption is also motivated if we model experts as learners rather than Bayesian agents: even if a forecaster starts out completely uninformed, they can quickly become calibrated in a domain simply by observing the frequency of events (Foster and Vohra, 1997).

Second, recent work has shown that modern deep neural networks are calibrated when trained on a proper loss function such as log loss. This is true for a variety of tasks, including image classification (Minderer et al., 2021; Hendrycks et al., 2020) and language modeling (Kadavath et al., 2022; Desai and Durrett, 2020; OpenAI, 2023); see (Blasiok et al., 2023) for a review of the literature. We may wish to use an ensemble of off-the-shelf neural networks for some prediction or classification task. If we trust these networks to be calibrated (as suggested by recent work), then we may wish to learn to ensemble these experts (models) in a way that has strong worst-case theoretical guarantees under the calibration assumption.

Logarithmic pooling is particularly sensible in the context of calibrated experts because it takes confident forecasts "more seriously" as compared with linear pooling (simple averaging). If Expert 1 reports probability distribution $(0.1\%, 99.9\%)$ over two outcomes and Expert 2 reports $(50\%, 50\%)$, then the logarithmic pool (with equal weights) is approximately $(3\%, 97\%)$, as compared with a linear pool of roughly $(25\%, 75\%)$. If Expert 1 is calibrated (as we are assuming), then the $(0.1\%, 99.9\%)$ forecast entails very strong evidence in favor of Outcome 2 over Outcome 1. Meanwhile, Expert 2's forecast gives no evidence either way. Thus, it is sensible for the aggregate to point to Outcome 2 over Outcome 1 with a fair amount of confidence.

As another example, suppose that Expert 1 reports $(0.04\%, 49.98\%, 49.98\%)$ and Expert 2 reports $(49.98\%, 0.04\%, 49.98\%)$ (a natural interpretation: Expert 1 found strong evidence against Outcome 1 and Expert 2 found strong evidence against Outcome 2). If both experts are calibrated, a sensible aggregate should arguably assign nearly all probability to Outcome 3. Logarithmic pooling returns roughly $(2.7\%, 2.7\%, 94.6\%)$, which (unlike linear pooling) accomplishes this.

Since we are allowing our algorithm to learn the optimal logarithmic pool, perhaps there is hope to compete not just with the best expert in hindsight, but the optimally weighted logarithmic pool of experts in hindsight. We will aim to compete with this stronger benchmark.

This paper demonstrates that the "calibrated experts" condition allows us to prove regret bounds when no such bounds are possible for an unrestricted adversary. While that is our primary motivation, the relaxation may also be of independent interest. For example, even in settings where vanishing regret is attainable in the presence of an unrestricted adversary, even stronger regret bounds might be achievable if calibration is assumed.

## 1.4 Our main result

Is vanishing regret possible in our setting? Our main result is that the answer is yes. We exhibit an algorithm that attains expected regret that scales as $O(\sqrt{T}\log T)$ with the number of time steps $T$. Our algorithm uses online mirror descent (OMD) with the *Tsallis entropy regularizer* $R(\mathbf{w}) = \frac{-1}{\alpha}(w_1^\alpha + \cdots + w_m^\alpha)$ and step size $\eta \approx \frac{1}{\sqrt{T}\ln T}$, where any choice of $\alpha \in (0, 1/2)$ attains the regret bound.

Our proof has two key ideas. One is to use the calibration property to show that the gradient of loss with respect to the weight vector is likely to be small (Section 4.4). This is how we leverage the calibration property to turn an intractable setting into one where — despite the unboundedness of log loss and the behavior of logarithmic pooling — there is hope for vanishing regret.

The other key idea (Section 4.3) involves keeping track of a function that, roughly speaking, reflects how much "regret potential" the algorithm has. We show that if the aforementioned gradient updates are indeed small, then this potential function decreases in value at each time step. This allows us to upper bound the algorithm's regret by the initial value of the potential function.

This potential argument is an important component of the proof. A naïve analysis might seek to use our bounds on the gradient steps to myopically bound the contribution to regret at each time step. Such an analysis, however, does not achieve our $O(\sqrt{T}\log T)$ regret bound. In particular, an adversary can force a large accumulation of regret if some experts' weights are very small (specifically by making the experts with small weights more informed than those with large weights) — but by doing so, the small weights increase and the adversary "spends down" its potential. Tracking this potential allows us to take this nuance into consideration, improving our bound.

We extend our main result by showing that the result holds even if experts are only approximately calibrated: so long as no expert understates the probability of an outcome by more than a constant factor, we still attain the same regret bound (see Corollary A.8). We also show in Appendix B that no OMD algorithm with a constant step size can attain expected regret better than $\Omega(\sqrt{T})$.

## 2 Related work

### 2.1 Probabilistic opinion pooling

There has been substantial mathematical work on probabilistic opinion pooling (i.e. forecast aggregation) since the 1980s. One line of work is axiomatic in nature: motivating opinion pooling methods by describing axioms that they satisfy. For example, logarithmic pooling satisfies unanimity preservation and external Bayesianality (Genest, 1984). There has also been work on Bayesian approaches to pooling, e.g. under the assumption that experts' signals are drawn from some parameterized class of distributions (Winkler, 1981; Lichtendahl et al., 2017).

Neyman and Roughgarden (2023) show that every proper loss function has an associated pooling method (the QA pool with respect to the loss function), which is the forecast that guarantees the largest possible overperformance (as judged by the loss function) compared with the strategy of choosing a random expert to trust. This mapping is a bijection: a pooling method is QA pooling with respect to some proper loss function if and only if it satisfies certain natural axioms.

### 2.2 Online prediction with expert advice

In the subfield of *prediction with expert advice*, for $T$ time steps, experts report "predictions" from a decision space $\mathcal{D}$ (often, as in our case, the space of probability distributions over a set of outcomes). A forecaster must then output their own prediction from $\mathcal{D}$. Then, predictions are assessed according to a loss function. See Cesa-Bianchi and Lugosi (2006) for an survey of this field.

We are particularly interested in *mixture forecasters*: forecasters who, instead of choosing an expert to trust at each time step, aggregate the expert' reports. Linear mixtures, i.e. convex combinations of predictions, have been especially well-studied, generally with the goal of learning weights for the convex combination to compete with the best weights in hindsight. Standard convex optimization algorithms achieve $O(\sqrt{T})$ regret for bounded, convex losses, but it is sometimes possible to do

better. For example, if the loss function is bounded and exp-concave, then logarithmic regret in $T$ is attainable (Cesa-Bianchi and Lugosi, 2006, §3.3).

Portfolio theory studies optimal stock selection for maximizing return on investment, often in a no-regret setting. Cover (1991) introduced the "universal portfolio" algorithm, which, for each of $T$ time steps, selects a portfolio (convex combination of stocks). Our setting translates naturally to Cover's: experts play the role of stocks, and the return of a stock corresponds to the probability that and expert assigns to the eventual outcome. The universal portfolio algorithm achieves logarithmic regret compared with the best portfolio in hindsight (Cover and Ordentlich, 1996); in our terms, this means that logarithmic regret (for log loss) is attainable for the linear pooling of experts. We refer the reader to (Li and Hoi, 2014) for a survey of this area.

To our knowledge, learning weights for *logarithmic* pooling has not been previously studied. As shown in Example 1.1, it is not possible to achieve vanishing regret if the setting is fully adversarial. We relax our setting by insisting that the experts be calibrated (see Section 3.1). To our knowledge, online prediction with expert advice has also not previously been studied under this condition.

## 3 Preliminaries

### 3.1 Calibration property

We define calibration as follows. Note that the definition is in the context of our setting, i.e. $m$ experts reporting probability distributions $\mathbf{p}^1, \ldots, \mathbf{p}^m$ over $n$ outcomes. We will use $J$ to denote the random variable corresponding to the outcome, i.e. $J$ takes values in $[n]$.

**Definition 3.1.** Consider a joint probability distribution $\mathbb{P}$ over experts' reports and the outcome.[3] We say that expert $i$ is *calibrated* if for all $\mathbf{p} \in \Delta^n$ and $j \in [n]$, we have that

$$\mathbb{P}\left[J = j \mid \mathbf{p}^i = \mathbf{p}\right] = p_j.$$

That is, expert $i$ is calibrated if the probability distribution of $J$ conditional on their report $\mathbf{p}^i$ is precisely $\mathbf{p}^i$. We say that $\mathbb{P}$ satisfies the *calibration property* if every expert is calibrated.

The key intuition behind the usefulness of calibration is that if an expert claims that an outcome is very unlikely, this is strong evidence that the outcome is in fact unlikely. In Section 4.4 we will use the calibration property to show that the gradient of the loss with respect to the weight vector is likely to be relatively small at each time step.

### 3.2 Our online learning setting

The setting for our online learning problem is as follows. For each time step $t \in [T]$:

(1) Our algorithm reports a weight vector $\mathbf{w}^t \in \Delta^m$.

(2) An adversary (with knowledge of $\mathbf{w}^t$) constructs a probability distribution $\mathbb{P}$, over reports and the outcome, that satisfies the calibration property.

(3) Reports $\mathbf{p}^{t,1}, \ldots, \mathbf{p}^{t,m}$ and an outcome $j$ are sampled from $\mathbb{P}$.

(4) The *loss* of a weight vector $\mathbf{w}$ is defined as $L^t(\mathbf{w}) := -\ln(p_j^*(\mathbf{w}))$, the log loss of the logarithmic pool of $\mathbf{p}^{t,1}, \ldots, \mathbf{p}^{t,m}$ with weights $\mathbf{w}$. Our algorithm incurs loss $L^t(\mathbf{w}^t)$.

We define the *regret* of our algorithm as

$$\text{Regret} = \sum_{t=1}^{T} L^t(\mathbf{w}^t) - \min_{\mathbf{w} \in \Delta^m} \sum_{t=1}^{T} L^t(\mathbf{w}).$$

That is, the benchmark for regret is the best weight vector in hindsight. Since our setting involves randomness, our goal is to provide an algorithm with vanishing *expected* regret against any adversarial strategy, where the expectation is taken over the sampling in Step (3).

---

[3]An equivalent, more mechanistic formulation views $\mathbb{P}$ instead as a joint probability distribution over *signals* received by each expert (i.e. the expert's private knowledge) and the outcome. Such a probability distribution is known as an *information structure*, see e.g. (Bergemann and Morris, 2019). Each expert computes their belief from their signal.

Even subject to the calibration property, the adversary has a large amount of flexibility, because the adversary retains control over the correlation between different experts' forecasts. An unrestricted adversary has exponentially many degrees of freedom (as a function of the number of experts), whereas the calibration property imposes a mere linear number of constraints.[4]

## 3.3 Our algorithm

We use Algorithm 1 to accomplish this goal. The algorithm is online mirror descent (OMD) on the weight vector. Fix any $\alpha \in (0, 1/2)$. We use the regularizer

$$R(\mathbf{w}) := \frac{-1}{\alpha}(w_1^\alpha + \cdots + w_m^\alpha).$$

This is known as the Tsallis entropy regularizer; see e.g. (Zimmert and Seldin, 2021) for previous use in the online learning literature. We obtain the same result (up to a multiplicative factor that depends on $\alpha$) regardless of the choice of $\alpha$. Because no choice of $\alpha$ stands out, we prove our result for all $\alpha \in (0, 1/2)$ simultaneously.

We will generally use a step size $\eta = \frac{1}{\sqrt{T}\ln T} \cdot \frac{1}{12m^{(1+\alpha)/2}n}$. However, in the (unlikely, as we show) event that some expert's weight becomes unusually small, we will reduce the step size.

---

**ALGORITHM 1:** OMD algorithm for learning weights for logarithmic pooling

$R(\mathbf{w}) := \frac{-1}{\alpha}(w_1^\alpha + \cdots + w_m^\alpha)$ ;        /* Any $\alpha \in (0, 1/2)$ will work */
$\eta \leftarrow \frac{1}{\sqrt{T}\ln T} \cdot \frac{1}{12m^{(1+\alpha)/2}n}$;
$\mathbf{w}^1 \leftarrow (1/m, \ldots, 1/m)$;
**for** $t = 1$ *to* $T$ **do**
    **if** $\eta \le \min_i((w_i^t)^\alpha)$ **then**
        $\eta_t \leftarrow \min(\eta_{t-1}, \eta)$;
    **else**
        $\eta_t \leftarrow \min(\eta_{t-1}, \min_i w_i^t)$ ;    /* Edge case; happens with low probability */
    **end**
    Observe loss function $L^t$ ;     /* $L^t$ is chosen as described in Section 3.2 */
    Define $\mathbf{w}^{t+1}$ such that $\nabla R(\mathbf{w}^{t+1}) = \nabla R(\mathbf{w}^t) - \eta_t \nabla L^t(\mathbf{w}^t)$;
**end**

---

In Appendix A, we prove that Algorithm 1 is efficient, taking $O(mn)$ time per time step.

Theorem 3.2 formally states our no-regret guarantee.

**Theorem 3.2.** *For any adversarial strategy, the expected regret[5] of Algorithm 1 is at most*

$$O\left(m^{(3-\alpha)/2}n\sqrt{T}\log T\right).$$

## 4 Proof of no-regret guarantee

In this section, we prove Theorem 3.2.

### 4.1 Outline of proof

We use the following fact, which follows from a more general statement about how losses relate to their associated quasi-arithmetic pools.

---

[4]This follows from the perspective of the adversary choosing an information structure from which experts' signals are drawn, as also alluded to in the previous footnote. The information structure specifies the probability of every possible combination of signals received by the experts, and thus has dimension that is exponential in the number of experts. The calibration property imposes linearly many constraints on this space.

[5]The given asymptotics assume that $T \gg m, n$, i.e. ignore terms that are lower-order in $T$.

**Proposition 4.1** (Follows from (Neyman and Roughgarden, 2023, Theorem 5.1)). *Let* $\mathbf{p}^1, \ldots, \mathbf{p}^m$ *be forecasts over $n$ outcomes, $j \in [n]$ be an outcome, and $\mathbf{w} \in \Delta^m$ be a weight vector. Let $\mathbf{p}^*(\mathbf{w})$ be the logarithmic pool of the forecasts with weight vector $\mathbf{w}$ and let $L(\mathbf{w}) := -\ln(p_j^*(\mathbf{w}))$ be the log loss of $\mathbf{p}^*(\mathbf{w})$ if Outcome $j$ is realized. Then $L$ is a convex function.*

In particular, all of our loss functions $L^t$ are convex, which means that standard regret bounds apply. In particular, to bound the expected regret of Algorithm 1, we will use a well-known regret bound for follow the regularized leader (FTRL) with linearized losses (Hazan, 2021, Lemma 5.3), which in our case is equivalent to OMD.[6]

**Lemma 4.2** (Follows from (Hazan, 2021, Lemma 5.3)). *If $\eta_t = \eta$ for all $t$, the regret of Algorithm 1 is at most*

$$\frac{1}{\eta} \left( \max_{\mathbf{w} \in \Delta^m} R(\mathbf{w}) - \min_{\mathbf{w} \in \Delta^m} R(\mathbf{w}) \right) + \sum_{t=1}^{T} \nabla L^t(\mathbf{w}^t) \cdot (\mathbf{w}^t - \mathbf{w}^{t+1}).$$

Informally, this bound means that if the vectors $\nabla L^t(\mathbf{w}^t)$ are small in magnitude, our regret is also small. Conversely, if some $\nabla L^t(\mathbf{w}^t)$ is large, this may be bad for our regret bound. We expect the gradient of the loss to be large if some expert is very wrong (assigns a very low probability to the correct outcome), since the loss would then be steeply increasing as a function of that expert's weight. Fortunately, the calibration property guarantees this to be unlikely. Specifically, we define the *small gradient assumption* as follows.

**Definition 4.3.** Define $\gamma := 12n \ln T$. The *small gradient assumption* holds for a particular run of Algorithm 1 if for every $t \in [T]$ and $i \in [m]$, we have

$$-\frac{\gamma}{w_i^t} \leq \partial_i L^t(\mathbf{w}^t) \leq \gamma,$$

where $\partial_i$ denotes the partial derivative with respect to the $i$-th weight.[7]

In Section 4.4, we prove that the small gradient assumption is very likely to hold. This is a key conceptual step in our proof, as it is where we leverage the calibration property to prove bounds that ultimately let us bound our algorithm's regret. We then use the low likelihood of the small gradient assumption failing in order to bound the contribution to the expected regret from the case where the assumption fails to hold.

In Sections 4.2 and 4.3, we bound regret under the condition that the small gradient assumption holds. We show that under the assumption, for all $i, t$ we have $(w_i^t)^\alpha \geq \eta$. Consequently, $\eta = \frac{1}{\sqrt{T} \ln T} \cdot \frac{1}{12m^{(1+\alpha)/2}n}$ at all time steps, so we can apply Lemma 4.2. The first term in the bound is $O(1/\eta) = O(\sqrt{T} \log T)$. As for the summation term, we upper bound it by keeping track of the following quantity:

$$\varphi(t) := \sum_{s=1}^{t} \nabla L^s(\mathbf{w}^s) \cdot (\mathbf{w}^s - \mathbf{w}^{s+1}) + 19m^2\gamma^2\eta(T - t) - 4\gamma \sum_{i=1}^{m} \ln w_i^{t+1}.$$

The first term is exactly the summation in Lemma 4.2 up through step $t$. The $19m^2\gamma^2\eta$ is something akin to an upper bound on the value of $L^t(\mathbf{w}^t) \cdot (\mathbf{w}^t - \mathbf{w}^{t+1})$ at a given time step (times $T - t$ remaining time steps). This upper bound is not strict: in particular, large summands are possible when some weights are small (because of the fact that the lower bound in the small gradient assumption is inversely proportional to $w_i^t$). However, attaining a large summand requires these small weights to increase, thus "spending potential" for future large summands. The last term keeps track of this potential.

We show that under the small gradient assumption, $\varphi(t)$ necessarily decreases with $t$. This argument, which we give in Section 4.3, is another key conceptual step, and is arguably the heart of the proof. Since $\varphi(T)$ is equal to the summation term in Lemma 4.2 (plus a positive number), and $\varphi(0) \geq \varphi(T)$, the summation term is less than or equal to $\varphi(0)$, which is at most $O(m^{(3-\alpha)/2}\sqrt{T} \log T)$. This completes the proof.

---

[6]This equivalence is due to our choice of regularizer, as we never need to project $\mathbf{w}^t$.

[7]See Equation 8 for an expression of this quantity in terms of the experts' reports and weights.

## 4.2 Bounds on $\mathrm{w}^t$ under the small gradient assumption

In this section, we state bounds on expert weights and how quickly they change from one time step to the next, conditional on the small gradient assumption. We use the following lemma, whose proof we defer to Appendix A.

**Lemma 4.4.** *Consider a particular run of Algorithm 1. Let $\zeta$ be a constant such that $-\frac{\zeta}{w_i^t} \leq \partial_i L^t(\mathbf{w}^t) \leq \zeta$ for all $i, t$. Then for every $i, t$, we have*

$$(w_i^t)^{\alpha-1} - \left( \frac{1}{w_i^t} + 1 \right) \eta_t \zeta \leq (w_i^{t+1})^{\alpha-1} \leq (w_i^t)^{\alpha-1} + \left( \frac{1}{\min_k w_k} + 1 \right) \eta_t \zeta.$$

*Furthermore, if $\eta_t \zeta \leq (1-\alpha)^2 (w_i^t)^\alpha$ for all $i$, then for every $i$ we have*

$$(w_i^{t+1})^{\alpha-1} \leq (w_i^t)^{\alpha-1} + (m+1)\eta_t \zeta.$$

Intuitively, this result states that when the gradient update is small, $w_i^{t+1}$ is not too different from $w_i^t$. Note that the lower bound $-\frac{\zeta}{w_i^t}$ that we place on the gradient is not a simple Lipschitz bound but instead depends on $w_i^t$; this makes the bounds in Lemma 4.4 less straightforward to prove. In particular, we bound each component $w_i^t$ individually, using bounds on the gradient of the loss for all other components and convexity arguments.

Lemma 4.4 can be translated into bounds on each $w_i^t$ and on the change between $w_i^t$ and $w_i^{t+1}$:

**Corollary 4.5.** *Under the small gradient assumption, for sufficiently large $T$ we have for all $i \in [m], t \in [T]$ that:*

*(#1) $(w_i^t)^\alpha \geq 4\eta\gamma$ and $w_i^t \geq \frac{1}{10\sqrt{m}} T^{1/(2(\alpha-1))}$.*

*(#2) $-32(w_i^t)^{1-\alpha}\eta\gamma \leq w_i^t - w_i^{t+1} \leq 2(w_i^t)^{2-\alpha}(m+1)\eta\gamma$.*

We defer the proof of Corollary 4.5 to Appendix A. The key idea for (#1) is to proceed by induction on $t$ on the two sub-statements in parallel: so long as $(w_i^t)^\alpha \geq 4\eta\gamma$, we may use the second part of Lemma 4.4 with $\zeta = \gamma$ to bound $(w_i^{t+1})^{\alpha-1}$ in terms of $(w_i^t)^{\alpha-1}$, which we can leverage to prove both sub-statements for $t + 1$. (#2) then follows from (#1) by routine (though nontrivial) algebra.

Armed with the bounds of Corollary 4.5, we are now able to show that under the small gradient assumption, Algorithm 1 attains vanishing regret.

## 4.3 Bounding regret under the small gradient assumption

Assume the small gradient assumption. Note that since $4\gamma \geq 1$, by Corollary 4.5 (#1) we have that $\eta_t = \eta$ for all $t$. This means that we may apply the bound in Lemma 4.2, and in particular we have

$$\frac{1}{\eta} \left( \max_{\mathbf{w} \in \Delta^m} R(\mathbf{w}) - \min_{\mathbf{w} \in \Delta^m} R(\mathbf{w}) \right) = \frac{1}{\eta} \cdot \frac{m}{\alpha} \left( \frac{1}{m} \right)^\alpha = \frac{m^{1-\alpha}}{\alpha\eta} = \frac{12}{\alpha} m^{(3-\alpha)/2} n \sqrt{T} \ln T.$$

It remains to bound the summation component of the regret bound in Lemma 4.2. To do so, we prove the following lemma, which we alluded to in Section 4.1 as the heart of the proof of Theorem 3.2.

**Lemma 4.6.** *For $t \in \{0, 1, \ldots, T\}$, let*

$$\varphi(t) := \sum_{s=1}^{t} \nabla L^s(\mathbf{w}^s) \cdot (\mathbf{w}^s - \mathbf{w}^{s+1}) + 19m^2\gamma^2\eta(T-t) - 4\gamma \sum_{i=1}^{m} \ln w_i^{t+1}.$$

*Under the small gradient assumption, for sufficiently large $T$, $\varphi(t)$ is a decreasing function of $t$.*

To prove this claim, consider a particular $t \in [T]$. We may write

$$\varphi(t) - \varphi(t-1) = \sum_{i=1}^{m} \left( (w_i^t - w_i^{t+1})\partial_i L^t(\mathbf{w}^t) - 19m\gamma^2\eta + 4\gamma(\ln w_i^t - \ln w_i^{t+1}) \right) \quad (1)$$

and we wish to show that this quantity is negative. In fact, we show that the contribution from every $i \in [m]$ is negative. The key idea is to consider two cases: $w_i^{t+1} \leq w_i^t$ and $w_i^{t+1} \geq w_i^t$. In each case, Corollary 4.5 provides an upper bound on the magnitude of the difference between $w_i^t$ and $w_i^{t+1}$. If $w_i^{t+1} \leq w_i^t$ then the first and third terms in the summation are positive but small, and are dominated by the middle term. If $w_i^{t+1} \geq w_i^t$ then the first term may be quite large, because of the asymmetric bound in the small gradient assumption (and the consequently asymmetric bound in Corollary 4.5). However, in this case the contribution of the third term is very negative, enough to make the overall expression negative. In this sense, the third term keeps track of unspent potential for future regret, which gets "spent down" whenever a large amount of regret is realized (as measured by the first term).

We now prove formally that each term of the summation in Equation 1 is negative.

*Proof.* First assume that $w_i^t - w_i^{t+1} \leq 0$. Note that by combining (#1) and (#2) of Corollary 4.5, we have

$$w_i^{t+1} - w_i^t \leq 32(w_i^t)^{1-\alpha}\eta\gamma \leq 8w_i^t.$$

By the small gradient assumption we have that

$$(w_i^t - w_i^{t+1})\partial_i L^t(\mathbf{w}^t) \leq \frac{\gamma(w_i^{t+1} - w_i^t)}{w_i^t}.$$

On the other hand, we have

$$4\gamma(\ln w_i^t - \ln w_i^{t+1}) = -4\gamma \ln\left(1 + \frac{w_i^{t+1} - w_i^t}{w_i^t}\right) \leq \frac{-\gamma(w_i^{t+1} - w_i^t)}{w_i^t}$$

for $T$ large enough. (Here we use that $w_i^{t+1} - w_i^t \leq 8w_i^t$ and that $\ln(1+x) \geq \frac{x}{4}$ for $x \leq 8$.) Thus, the first and third terms in Equation 1 are net negative; meanwhile, the second term is also negative, so the expression is negative.

Now assume that $w_i^t - w_i^{t+1} \geq 0$. Again by the small gradient assumption, we have that

$$(w_i^t - w_i^{t+1})\partial_i L^t(\mathbf{w}^t) \leq \gamma(w_i^t - w_i^{t+1}) \leq 2(m+1)\eta\gamma^2(w_i^t)^{2-\alpha} \leq 3m\eta\gamma^2$$

and

$$4\gamma(\ln w_i^t - \ln w_i^{t+1}) = -4\gamma \ln\left(1 - \frac{w_i^t - w_i^{t+1}}{w_i^t}\right) \leq -4\gamma \ln(1 - 2(m+1)\eta\gamma(w_i^t)^{1-\alpha})$$

$$\leq -4\gamma \ln(1 - 2(m+1)\eta\gamma) \leq -4\gamma \ln(1 - 3m\eta\gamma) \leq 16m\eta\gamma^2$$

for $T$ sufficiently large, where in the last step we use that $\ln(1-x) \geq -\frac{4}{3}x$ for $x > 0$ sufficiently small (and we have $\lim_{T\to\infty} 3m\eta\gamma = 0$). Since $16 + 3 \leq 19$, the right-hand side of Equation 1 is negative. This concludes the proof. □

**Corollary 4.7.** *For sufficiently large $T$, under the small gradient assumption, the regret of Algorithm 1 is at most $\left(240 + \frac{12}{\alpha}\right) m^{(3-\alpha)/2} n\sqrt{T} \ln T$.*

*Proof.* We have already bounded the first term in the regret bound in Lemma 4.2. It remains only to bound the second term. This term is exactly equal to $\varphi(T) + 4\gamma \sum_{i=1}^m \ln w_i^{T+1} \leq \varphi(T)$, and $\varphi(T) \leq \varphi(0)$, by Lemma 4.6. We have

$$\varphi(0) = 19m^2\gamma^2\eta T + 4m\gamma \ln m \leq 20m^2\gamma^2\eta T$$

for sufficiently large $T$. Plugging in $\gamma = 12n \ln T$ and $\eta = \frac{1}{\sqrt{T} \ln T} \cdot \frac{1}{12m^{(1+\alpha)/2}n}$ concludes the proof. □

## 4.4 The case where the small gradient assumption fails

It remains to consider the case in which the small gradient assumption does not hold. This part of the proof consists primarily of technical lemmas, which we defer to Appendix A. The key lemma is a bound on the probability that the small gradient assumption fails by a given margin:

**Lemma 4.8.** *For any weight vector* $\mathbf{w}$, $i \in [m]$, *and* $\zeta \geq 0$, *we have that*

$$\mathbb{P}\left[\partial_i L(\mathbf{w}) \geq \zeta\right] \leq n e^{-\zeta} \tag{2}$$

*and*

$$\mathbb{P}\left[\partial_i L(\mathbf{w}) \leq -\frac{\zeta}{w_i}\right] \leq mn^2 e^{-\zeta/n}. \tag{3}$$

Note that plugging in $\zeta = \gamma$ yields a bound of $mT(ne^{-\gamma} + mn^2 e^{-\gamma/n})$ on the probability that the small gradient assumption fails to hold. (Since $\gamma = 12n \ln T$, this quantity is on the order of $T^{-11}$.)

The proof of Lemma 4.8 is the only part of the proof of Theorem 3.2 that uses the calibration property. While we defer the full proof to Appendix A, we highlight how the calibration property is used to prove Equation 2. In brief, it is straightforward to show that $\partial_i L(\mathbf{w}) \leq -\ln p_j^i$, where $J$ is the random variable corresponding to the realized outcome.[8] Therefore, we have

$$\mathbb{P}\left[\partial_i L(\mathbf{w}) \geq \zeta\right] \leq \mathbb{P}\left[-\ln p_J^i \geq \zeta\right] = \mathbb{P}\left[p_J^i \leq e^{-\zeta}\right] = \sum_{j=1}^n \mathbb{P}\left[J = j \;\&\; p_j^i \leq e^{-\zeta}\right]$$

$$= \sum_{j=1}^n \mathbb{P}\left[p_j^i \leq e^{-\zeta}\right] \mathbb{P}\left[J = j \mid p_j^i \leq e^{-\zeta}\right] \leq \sum_{j=1}^n \mathbb{P}\left[J = j \mid p_j^i \leq e^{-\zeta}\right] \leq n e^{-\zeta},$$

where the last step follows by the calibration property, thus proving Equation 2.

Combining Lemma 4.8 with an analysis of our algorithm using the standard regret bound for online mirror descent (Orabona, 2021, Theorem 6.8) gives us the following result as a corollary.

**Corollary 4.9.** *The expected total regret of our algorithm conditional on the small gradient assumption* not *holding, times the probability of this event, is at most* $\tilde{O}(T^{(5-\alpha)/(1-\alpha)-10})$.

It follows that the contribution to expected regret from the case that the small gradient assumption does not hold is $\tilde{O}(T^{-1})$, which is negligible. Together with Corollary 4.7 (which bounds regret under the small gradient assumption), this proves Theorem 3.2. As a matter of fact, Theorem 3.2 holds even if experts are only *approximately* calibrated. As with other details of this section, we refer the reader to Appendix A.

## 5 Conclusion

In this work we have considered the problem of learning optimal weights for the logarithmic pooling of expert forecasts. It quickly became apparent that under the usual fully adversarial setup, attaining vanishing regret is impossible (Example 1.1). We chose to relax the environment by imposing the constraint on the adversary that experts must be calibrated. Put otherwise, the adversary is allowed to choose a joint probability distribution over the experts' reports and the outcome however it wants to, so long as the experts' reports are calibrated, after which the realized reports and outcome are selected at random from this distribution. To our knowledge, this setting is a novel contribution to the literature on prediction with expert advice. The setting may be of independent interest: we have demonstrated that no-regret bounds are possible in this setting when they are otherwise impossible, and it seems plausible that even in settings where no-regret bounds are attainable in a fully adversarial setting, the calibration property allows for stronger results.

Another important direction for future work is learning weights for other pooling methods. In particular, because of the close connection between a proper loss function and its associated quasi-arithmetic pool, it is natural to ask for which proper loss functions it is possible to achieve vanishing regret when learning weights for quasi-arithmetic pooling with respect to the loss function. (Neyman and Roughgarden, 2023, §5) showed that the loss of a quasi-arithmetic pool is convex in the experts' weights, and therefore the usual no-regret algorithms (e.g. online gradient descent) guarantee $O(\sqrt{T})$ regret — so long as the loss function is bounded. In this work, we extended their result to the log loss (with the associated QA pooling method, i.e. logarithmic pooling.) Extending our techniques to other unbounded loss functions is a promising avenue for future exploration.

---

[8]Writing out the expression for $L(\mathbf{w})$ and differentiating leaves us with $-\ln p_j^i$ plus a negative term – see Equation 8.

## Acknowledgments and Disclosure of Funding

Funding in direct support of this work: NSF grant DGE-2036197; NSF grant CCF-2006737; and ARO grant W911NF1910294.

We would like to thank Christian Kroer, Rafael Frongillo, and Bo Waggoner for discussion.

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

## A  Omitted Proofs

**Efficiency of Algorithm 1**  The only nontrivial step of the algorithm is finding the weight vector satisfying the equation on the last line of the algorithm. To do so, it is first necessary to compute the gradient of the loss. This gradient, given by Equation 8 below, can clearly be computed in time $O(mn)$. After that, it is necessary to find the weight vector $\mathbf{w}^{t+1}$ that satisfies the equation on the last line. This can be done efficiently through local search: the goal amounts to find weights $(w_1, \ldots, w_m)$ such that the vector $(w_1^{\alpha-1}, \ldots, w_m^{\alpha-1})$ is equal to a target vector (call it $\mathbf{v}$) plus a constant $c$ times the all-ones vector. That is, we need to simultaneously solve the equation $w_i^{\alpha-1} = v_i + c$ for all $i$, with weights that add to 1. (Here, the $v_i$ are knowns and the $w_i$ and $c$ are unknowns.)

We start by finding $c$, by solving the equation $\sum_i (v_i + c)^{1/(\alpha-1)} = 1$. Such a $c$ exists because the left-hand side of this equation is continuous and monotone decreasing, going from infinity to zero as $c$ ranges from $-\min_i v_i$ to infinity. We can solve for $c$ very efficiently, e.g. with Newton's method. Once we know $c$, we know each $w_i$: we have $w_i = (v_i + c)^{1/(\alpha-1)}$. Thus, Algorithm 1 takes $O(mn)$ time.

*Proof of Lemma 4.4.*  Fix any $t$. Note that since the space of possible weights is $\Delta^m$, it is most natural to think of $\nabla R$ as a function from $\Delta^m$ to $\mathbb{R}^m / T(\mathbf{1}_m)$, i.e. $\mathbb{R}^m$ modulo translation by the all-ones vector (which is orthogonal to $\Delta^m$ in $\mathbb{R}^m$). That is, $\nabla R(\mathbf{w}) = -((w_1)^{\alpha-1}, \ldots, (w_m)^{\alpha-1})$, where this vector may be thought of as modulo translation by the all-ones vector. Nevertheless, we find it convenient to define $\partial_i R(\mathbf{w}) := -(w_i)^{\alpha-1}$. We define $\partial_i L^t(\mathbf{w})$ similarly (see Section 4.4).

Define $\mathbf{h} \in \mathbb{R}^m$ to have coordinates $h_i := \partial_i R(\mathbf{w}^t) - \eta_t \partial_i L^t(\mathbf{w}^t)$. Per the update rule, we have that $h_i \equiv R(\mathbf{w}^{t+1}) \mod T(\mathbf{1}_m)$. We have

$$-(w_i^t)^{\alpha-1} - \eta_t \zeta = \partial_i R(\mathbf{w}^t) - \eta_t \zeta \leq h_i \leq \partial_i R(\mathbf{w}^t) + \frac{\eta_t \zeta}{w_i^t} = -(w_i^t)^{\alpha-1} + \frac{\eta_t \zeta}{w_i^t} \qquad (4)$$

Applying the first and last claims of Lemma A.1 (below) with $a = \alpha - 1$, $\mathbf{v} = \mathbf{w}^t$, $\kappa = \eta_t \zeta$, and $\mathbf{g} = -\mathbf{h}$, we have that there exists a unique $c \in \mathbb{R}$ such that

$$\sum_{i=1}^m (-h_i + c)^{1/(\alpha-1)} = 1,$$

and in fact that $-\eta_t \zeta \leq c \leq m \eta_t \zeta$. (Equation 4 is relevant here because it is equivalent to the $v_i^a - \frac{\kappa}{w_i^t} \leq g_i \leq v_i^a + \kappa$ conditions in Lemma A.1. This is also where we use that $\eta_t \zeta \leq (1-\alpha)^2 (w_i^t)^\alpha$, which is equivalent to $\kappa \leq a^2 v_i^{a+1}$.) The significance of this fact is that $(-h_i + c)^{1/(\alpha-1)}$ is precisely $w_i^{t+1}$, since (in $\mathbb{R}^m$) we have that $\left(\partial_i R(\mathbf{w}^{t+1}), \ldots, \partial_i R(\mathbf{w}^{t+1})\right) = \mathbf{h} - c \cdot \mathbf{1}$ for *some* $c$, and in particular this $c$ must be such that $\sum_i w_i^{t+1} = 1$. In particular, this means that for all $i$, we have

$$(w_i^{t+1})^{\alpha-1} = -h_i + c \leq (w_i^t)^{\alpha-1} + \eta_t \zeta + \frac{1}{\min_k w_k} \eta_t \zeta = (w_i^t)^{\alpha-1} + \left(\frac{1}{\min_k w_k} + 1\right) \eta_t \zeta.$$

Here, the inequality comes from the left inequality of Equation 4 and the fact that $c \leq \frac{1}{\min_k w_k} \eta_t \zeta$. If we also have that $\eta_t \zeta \leq (1-\alpha)^2 (w_i^t)^\alpha$, then the last claim of Lemma A.1 gives us that

$$(w_i^{t+1})^{\alpha-1} = -h_i + c \leq (w_i^t)^{\alpha-1} + \eta_t \zeta + m \eta_t \zeta = (w_i^t)^{\alpha-1} + (m+1) \eta_t \zeta.$$

Similarly, we have

$$(w_i^{t+1})^{\alpha-1} = -h_i + c \geq (w_i^t)^{\alpha-1} - \frac{\eta_t \zeta}{w_i^t} - \eta_t \zeta = (w_i^t)^{\alpha-1} - \left(\frac{1}{w_i^t} + 1\right) \eta_t \zeta.$$

$\square$

**Lemma A.1.**  *Let $-1 < a < 0$ and $\mathbf{g} \in \mathbb{R}^m$. There is a unique $c \in \mathbb{R}$ such that $\sum_i (g_i + c)^{1/a} = 1$. Furthermore, let $\mathbf{v} \in \Delta^m$ and $\kappa \geq 0$. Then:*

- *If $g_i \leq v_i^a + \kappa$ for all $i$, then $c \geq -\kappa$.*

- *If $g_i \geq v_i^a - \frac{\kappa}{v_i}$ for all $i$, then $c \leq \frac{\kappa}{\min_i v_i}$.*

– *And if, furthermore, $\kappa \leq a^2 v_i^{a+1}$ for all $i$, then $c \leq m\kappa$.*

*Proof.* Observe that $\sum_i (g_i + c)^{1/a}$ is a continuous, monotone decreasing function on $c \in (-\min_i g_i, \infty)$; the range of the function on this interval is $(0, \infty)$. Therefore, there is a unique $c \in (-\min_i g_i, \infty)$ such that the sum equals 1.

We now prove the first bullet. Since $x^{1/a}$ decreases in $x$ and $g_i \leq v_i^a + \kappa$, we have that

$$1 = \sum_i (g_i + c)^{1/a} \geq \sum_i (v_i^a + \kappa + c)^{1/a}.$$

Suppose for contradiction that $c < -\kappa$. Then $v_i^a + \kappa + c < v_i^a$ for all $i$, so

$$\sum_i (v_i^a + \kappa + c)^{1/a} > \sum_i (v_i^a)^{1/a} = \sum_i v_i = 1.$$

This is a contradiction, so in fact $c \geq -\kappa$.

The first claim of the second bullet is analogous. Since $x^{1/a}$ decreases in $x$ and $g_i \geq v_i^a - \kappa v_i$, we have that

$$1 = \sum_i (g_i + c)^{1/a} \leq \sum_i \left( v_i^a - \frac{\kappa}{v_i} + c \right)^{1/a}. \tag{5}$$

Suppose for contradiction that $c > \frac{\kappa}{v_i}$ for every $i$. Then $v_i^a - \frac{\kappa}{v_i} + c > v_i^a$ for all $i$, so

$$\sum_i \left( v_i^a - \frac{\kappa}{v_i} + c \right)^{1/a} < \sum_i (v_i^a)^{1/a} = \sum_i v_i = 1.$$

This is a contradiction, so in fact $c \leq \frac{\kappa}{\min_i v_i}$.

We now prove the second claim of the second bullet. To do so, we note the following technical lemma (proof below).

**Lemma A.2.** *For $-1 < a < 0$ and $\kappa, c \geq 0$, the function $f(x) = \left( x^a - \frac{\kappa}{x} + c \right)^{1/a}$ is defined and concave at any value of $x > 0$ such that $a^2 x^{a+1} \geq \kappa$.*

Since for a general concave function $f$ it holds that $\frac{1}{m} \sum_{i=1}^m f(x_i) \leq f\left( \frac{1}{m} \sum_{i=1}^m x_i \right)$, the following inequality follows from Lemma A.2:

$$\sum_i \left( v_i^a - \frac{\kappa}{v_i} + c \right)^{1/a} \leq m \left( \left( \frac{1}{m} \right)^a - \kappa m + c \right)^{1/a}.$$

(Here we are using the fact that $\sum_i v_i = 1$.) Now, combining this fact with Equation 5, we have that

$$m \left( \left( \frac{1}{m} \right)^a - \kappa m + c \right)^{1/a} \geq 1$$

$$\left( \frac{1}{m} \right)^a - \kappa m + c \leq \left( \frac{1}{m} \right)^a$$

so $c \leq m\kappa$, as desired. $\square$

*Proof of Lemma A.2.* To show that $f$ is defined for any $x$ such that $a^2 x^{a+1} \geq \kappa$, we need to show that $x^a - \frac{\kappa}{x} + c > 0$ for such values of $x$. This is indeed the case:

$$x^a - \frac{\kappa}{x} + c \geq x^a - a^2 x^a + c = (1 - a^2)x^a + c > c \geq 0.$$

Now we show concavity. We have

$$f''(x) = \frac{1}{-a} \left( \left( 1 + \frac{1}{-a} \right) \left( x^a - \frac{\kappa}{x} + c \right)^{1/a-2} \left( ax^{a-1} + \frac{\kappa}{x^2} \right)^2 - \left( x^a - \frac{\kappa}{x} + c \right)^{1/a-1} \left( a(a-1)x^{a-2} - \frac{2\kappa}{x^3} \right) \right)$$

so we wish to show that

$$\left(1 + \frac{1}{-a}\right)\left(x^a - \frac{\kappa}{x} + c\right)^{1/a-2}\left(ax^{a-1} + \frac{\kappa}{x^2}\right)^2 \leq \left(x^a - \frac{\kappa}{x} + c\right)^{1/a-1}\left(a(a-1)x^{a-2} - \frac{2\kappa}{x^3}\right)$$

for every $x$ such that $a^2 x^{a+1} \geq \kappa$. Fix any such $x$, and let $d = \frac{\kappa}{x^{a+1}}$ (so $0 \leq d \leq a^2$). We have

$$d \leq a^2$$

$$(1+a)(a^2 - d)d \geq 0$$

$$(1-a)(a+d)^2 \leq -a(1-d)(a(a-1) - 2d) \qquad \text{(rearrange terms)}$$

$$\left(1 - \frac{1}{a}\right)(a+d)^2 x^a \leq ((1-d)x^a)(a(a-1) - 2d) \qquad \text{(multiply by } \frac{x^a}{-a}\text{)}$$

$$\left(1 - \frac{1}{a}\right)(a+d)^2 x^a \leq ((1-d)x^a + c)(a(a-1) - 2d) \qquad (c(a(a-1) - 2d) \geq 0)$$

$$\left(1 - \frac{1}{a}\right)((a+d)x^{a-1})^2 \leq ((1-d)x^a + c)(a(a-1) - 2d)x^{a-2} \qquad \text{(multiply by } x^{a-2}\text{)}$$

$$\left(1 - \frac{1}{a}\right)\left(ax^{a-1} + \frac{\kappa}{x^2}\right)^2 \leq \left(x^a - \frac{\kappa}{x} + c\right)\left(a(a-1)x^{a-2} - \frac{2\kappa}{x^3}\right) \qquad \text{(substitute } d = \kappa x^{-a-1}\text{)}.$$

Note that the fifth line is justified by the fact that $c \geq 0$ and $a(a-1) \geq 2d$ (because $a^2 \geq d$ and $-a > a^2 \geq d$). Now, multiplying both sides by $\left(x^a - \frac{\kappa}{x} + c\right)^{1/a-2}$ completes the proof. $\qquad\square$

*Proof of Corollary 4.5.* Note that $\eta\gamma = \frac{1}{\sqrt{T}m^{(1+\alpha)/2}}$ and also that $\eta_t \leq \eta$ for all $t$; we will be using these facts.

To prove (#1), we proceed by induction on $t$. In the case of $t = 1$, all weights are $1/m$, so the claim holds for sufficiently large $T$. Now assume that the claim holds for a generic $t < T$; we show it for $t + 1$.

By the small gradient assumption, we may use Lemma 4.4 with $\zeta = \gamma$. By the inductive hypothesis (and the fact that $\eta_t \leq \eta$), we may apply the second part of Lemma 4.4:

$$(w_i^{t+1})^{\alpha-1} \leq (w_i^t)^{\alpha-1} + (m+1)\eta\gamma \leq \cdots \leq (1/m)^{\alpha-1} + t(m+1)\eta\gamma.$$

$$\leq (1/m)^{\alpha-1} + \frac{(T-1)(m+1)}{m^{(1+\alpha)/2}\sqrt{T}} \leq 3m^{(1-\alpha)/2}\sqrt{T}.$$

Since $\frac{-1}{2} < \alpha - 1 < 0$, this means that $w_i^t \geq \frac{1}{10\sqrt{m}}T^{1/(2(\alpha-1))}$.

We also have that

$$(w_i^{t+1})^\alpha \geq \frac{1}{(10\sqrt{m})^\alpha}T^{\alpha/(2(\alpha-1))} \geq \frac{4}{m^{(1+\alpha)/2}}T^{-1/2} = 4\eta\gamma$$

for $T$ sufficiently large, since $\frac{\alpha}{2(\alpha-1)} > \frac{-1}{2}$. This completes the inductive step, and thus the proof of (#1).

To prove (#2), we use the following technical lemma (see below for the proof).

**Lemma A.3.** *Fix $x > 0$ and $-1 < a < 0$. Let $f(y) = (x^a + y)^{1/a}$. Then for all $y > -x^a$, we have*

$$x - f(y) \leq \frac{-1}{a}x^{1-a}y \tag{6}$$

*and for all $-1 < c \leq 0$, for all $cx^a \leq y \leq 0$, we have*

$$f(y) - x \leq \frac{1}{a}(1+c)^{1/a-1}x^{1-a}y. \tag{7}$$

We apply Equation 6 to $x = w_i^t$, $y = (m+1)\eta\gamma$, and $a = \alpha - 1$. This tells us that

$$w_i^t - w_i^{t+1} \leq w_i^t - ((w_i^t)^{\alpha-1} + (m+1)\eta\gamma)^{1/(\alpha-1)} \leq 2(w_i^t)^{2-\alpha}(m+1)\eta\gamma.$$

The first step follows by the second part of Lemma 4.4 and the fact that $\eta_t \leq \eta$. The second step follows from Equation 6 and uses the fact that $\frac{1}{1-\alpha} > 2$.

For the other side of (#2), we observe that since by (#1) we have $(w_i^t)^\alpha \geq 4\eta\gamma$, it follows that $\frac{1}{2}(w_i^t)^\alpha \geq (w_i^t + 1)\eta\gamma$, and so $\left(\frac{1}{w_i^t} + 1\right)\eta\gamma \leq \frac{1}{2}(w_i^t)^{\alpha-1}$. Therefore, we can apply Equation 7 to $x = w_i^t$, $y = -\left(\frac{1}{w_i^t} + 1\right)\eta\gamma$, $a = \alpha - 1$, and $c = -\frac{1}{2}$. This tells us that

$$w_i^{t+1} - w_i^t \leq \left((w_i^t)^{\alpha-1} - \left(\frac{1}{w_i^t} + 1\right)\eta\gamma\right)^{1/(\alpha-1)} - w_i^t \leq 16(w_i^t)^{2-\alpha}\left(\frac{1}{w_i^t} + 1\right)\eta\gamma$$
$$\leq 32(w_i^t)^{1-\alpha}\eta\gamma.$$

This completes the proof. $\qquad\square$

*Proof of Lemma A.3.* For all $y > -x^a$, we have

$$f'(y) = \frac{1}{a}(x^a + y)^{1/a-1}$$

and

$$f''(y) = \frac{1}{a}\left(\frac{1}{a} - 1\right)(x^a + y)^{1/a-2} > 0,$$

so $f'$ is increasing. Thus, for positive values of $y$ we have

$$f'(0) \leq \frac{f(y) - f(0)}{y} = \frac{f(y) - x}{y} \leq f'(y)$$

and for negative values of $y$ we have

$$f'(y) \leq \frac{f(y) - f(0)}{y} = \frac{f(y) - x}{y} \leq f'(0).$$

Regardless of whether $y$ is positive or negative, this means that $x - f(y) \leq -yf'(0) = \frac{-1}{a}x^{1-a}y$.

Now, let $-1 < c \leq 0$ and suppose that $cx^a \leq y \leq 0$. Since $f'$ is increasing, we have that

$$f'(y) \geq f'(cx^a) = \frac{1}{a}((1+c)x^a)^{1/a-1} = \frac{1}{a}(1+c)^{1/a-1}x^{1-a},$$

so

$$f(y) - x \leq yf'(y) \leq \frac{1}{a}(1+c)^{1/a-1}x^{1-a}y.$$

$\qquad\square$

*Proof of Lemma 4.8.* We first derive an expression for $\partial_i L(\mathbf{w})$ given expert reports $\mathbf{p}^1, \ldots, \mathbf{p}^m$, where $L(\mathbf{w})$ is the log loss of the logarithmic pool $\mathbf{p}^*(\mathbf{w})$ of $\mathbf{p}^1, \ldots, \mathbf{p}^m$ with weights $\mathbf{w}$, and $j$ is the realized outcome. We have[9]

$$\partial_i L(\mathbf{w}) = -\partial_i \ln \frac{\prod_{k=1}^m (p_j^k)^{w_k}}{\sum_{\ell=1}^n \prod_{k=1}^m (p_\ell^k)^{w_k}} = \partial_i \ln \left(\sum_{\ell=1}^n \prod_{k=1}^m (p_\ell^k)^{w_k}\right) - \partial_i \ln \left(\prod_{k=1}^m (p_j^k)^{w_k}\right)$$
$$= \frac{\sum_{\ell=1}^n \ln p_\ell^i \cdot \prod_{k=1}^m (p_\ell^k)^{w_k}}{\sum_{\ell=1}^n \prod_{k=1}^m (p_\ell^k)^{w_k}} - \ln p_j^i = \sum_{\ell=1}^n p_\ell^*(\mathbf{w}) \ln p_\ell^i - \ln p_j^i. \qquad (8)$$

---

[9]It should be noted that $\nabla L(\mathbf{w})$ is most naturally thought of as living in $\mathbb{R}^m/T(\mathbf{1}_m)$, i.e. $m$-dimensional space modulo translation by the all-ones vector, since $\mathbf{w}$ lives in a place that is orthogonal to the all-ones vector. As an arbitrary but convenient convention, we define $\partial_i L(\mathbf{w})$ to be the specific value derived below, and define the small gradient assumption accordingly.

Equation (2) now follows fairly straightforwardly. Equation 8 tells us that $\partial_i L(\mathbf{w}) \leq -\ln p_J^i$, where $J$ is the random variable corresponding to the realized outcome. Therefore, we have

$$\mathbb{P}\left[\partial_i L(\mathbf{w}) \geq \zeta\right] \leq \mathbb{P}\left[-\ln p_J^i \geq \zeta\right] = \mathbb{P}\left[p_J^i \leq e^{-\zeta}\right] = \sum_{j=1}^{n} \mathbb{P}\left[J = j \ \& \ p_j^i \leq e^{-\zeta}\right]$$

$$= \sum_{j=1}^{n} \mathbb{P}\left[p_j^i \leq e^{-\zeta}\right] \mathbb{P}\left[J = j \mid p_j^i \leq e^{-\zeta}\right] \leq \sum_{j=1}^{n} \mathbb{P}\left[J = j \mid p_j^i \leq e^{-\zeta}\right] \leq ne^{-\zeta},$$

where the last step follows by the calibration property. This proves Equation (2).

We now prove Equation (3). The proof has a similar idea, but is somewhat more technical. We begin by proving the following lemma; we again use the calibration property in the proof.

**Lemma A.4.** *For all $q$, we have*

$$\mathbb{P}\left[\forall j \exists i : p_j^i \leq q\right] \leq mnq.$$

*Proof.* Let $J$ be the random variable corresponding to the index of the outcome that ends up happening. We have

$$\mathbb{P}\left[\forall j \exists i : p_j^i \leq q\right] \leq \mathbb{P}\left[\exists i : p_J^i \leq q\right] = \sum_{j \in [n]} \mathbb{P}\left[J = j \ \& \ \exists i : p_j^i \leq q\right]$$

$$\leq \sum_{j \in [n]} \sum_{i \in [m]} \mathbb{P}\left[J = j \ \& \ p_j^i \leq q\right]$$

$$= \sum_{j \in [n]} \sum_{i \in [m]} \mathbb{P}\left[p_j^i \leq q\right] \mathbb{P}\left[J = j \mid p_j^i \leq q\right] \leq \sum_{j \in [n]} \sum_{i \in [m]} 1 \cdot q = mnq,$$

where the fact that $\mathbb{P}\left[J = j \mid p_j^i \leq q\right] \leq q$ follows by the calibration property. $\square$

**Corollary A.5.** *For any reports $\mathbf{p}^1, \ldots, \mathbf{p}^m$, weight vector $\mathbf{w}$, $i \in [m]$, and $j \in [n]$, we have*

$$\mathbb{P}\left[p_j^*(\mathbf{w}) \geq \frac{(p_j^i)^{w_i}}{q}\right] \leq mnq.$$

*Proof.* We have

$$p_j^*(\mathbf{w}) = \frac{\prod_{k=1}^{m}(p_j^k)^{w_k}}{\sum_{\ell=1}^{n}\prod_{k=1}^{m}(p_\ell^k)^{w_k}} \leq \frac{(p_j^i)^{w_i}}{\sum_{\ell=1}^{n}\prod_{k=1}^{m}(p_\ell^k)^{w_k}}.$$

Now, assuming that there is an $\ell$ such that for every $k$ we have $p_\ell^k > q$, the denominator is greater than $q$, in which case we have $p_j^*(\mathbf{w}) < \frac{(p_j^i)^{w_i}}{q}$. Therefore, if $p_j^*(\mathbf{w}) \geq \frac{(p_j^i)^{w_i}}{q}$, it follows that for every $\ell$ there is a $k$ such that $p_\ell^k \leq q$. By Lemma A.4, this happens with probability at most $mnq$. $\square$

We now use Corollary A.5 to prove Equation (3). Note that the equation is trivial for $\zeta < n$, so we assume that $\zeta \geq n$. By setting $q := e^{\frac{-\zeta}{n}}$, we may restate Equation (3) as follows: for any $q \leq \frac{1}{e}$, any $i \in [m]$, and any weight vector $\mathbf{w}$,

$$\mathbb{P}\left[\partial_i L(\mathbf{w}) \leq -\frac{n \ln 1/q}{w_i}\right] \leq mn^2 q.$$

(Note that the condition $q \leq \frac{1}{e}$ is equivalent to $\zeta \geq n$.) We prove this result.

From Equation 8, we have

$$\partial_i L(\mathbf{w}) = \sum_{j=1}^{n} p_j^*(\mathbf{w}) \ln p_j^i - \ln p_j^i \geq \sum_{j=1}^{n} p_j(\mathbf{w}) \ln p_j^i.$$

Now, it suffices to show that for each $j \in [n]$, the probability that $p_j(\mathbf{w}) \ln p_j^i \leq -\frac{\ln 1/q}{w_i} = \frac{\ln q}{w_i}$ is at most $mnq$; the desired result will then follow by the union bound. By Corollary A.5, for each $j$ we have that

$$\mathbb{P}\left[p_j(\mathbf{w}) \ln p_j^i \leq \frac{(p_j^i)^{w_i}}{q} \ln p_j^i\right] \leq mnq.$$

Additionally, we know for a fact that $p_j(\mathbf{w}) \ln p_j^i \geq \ln p_j^i$ (since $p_j(\mathbf{w}) \leq 1$), so in fact

$$\mathbb{P}\left[p_j(\mathbf{w}) \ln p_j^i \leq \max\left(\frac{(p_j^i)^{w_i}}{q} \ln p_j^i, \ln p_j^i\right)\right] \leq mnq.$$

It remains only to show that $\max\left(\frac{(p_j^i)^{w_i}}{q} \ln p_j^i, \ln p_j^i\right) \geq \frac{\ln q}{w_i}$. If $p_j^i \geq q^{1/w_i}$ then this is clearly true, since in that case $\ln p_j^i \geq \frac{\ln q}{w_i}$. Now suppose that $p_j^i < q^{1/w_i}$. Observe that $\frac{x^{w_i}}{q} \ln x$ decreases on $(0, e^{-1/w_i})$, and that (since $q \leq \frac{1}{e}$) we have $q^{1/w_i} \leq e^{-1/w_i}$. Therefore,

$$\frac{(p_j^i)^{w_i}}{q} \ln p_j^i \leq \frac{(q^{1/w_i})^{w_i}}{q} \ln q^{1/w_i} = \frac{\ln q}{w_i}.$$

This completes the proof of Equation (3), and thus of Lemma 4.8. $\qquad\square$

The following lemma lower bounds the regret of Algorithm 1 as a function of $\zeta$.

**Lemma A.6.** *Consider a run of Algorithm 1. Let $\zeta$ be such that $-\frac{\zeta}{w_i^t} \leq \partial_i L^t(\mathbf{w}^t) \leq \zeta$ for all $i, t$. The total regret is at most*

$$O\left(\zeta^{2(2-\alpha)/(1-\alpha)} T^{(5-\alpha)/(1-\alpha)}\right).$$

*Proof of Lemma A.6.* We first bound $w_i^t$ for all $i, t$. From Lemma 4.4, we have that

$$(w_i^{t+1})^{\alpha-1} \leq (w_i^t)^{\alpha-1} + \left(\frac{1}{\min_i w_i^t} + 1\right) \eta_t \zeta \leq (w_i^t)^{\alpha-1} + 2\zeta.$$

Here we use that $\frac{1}{\min_i w_i} + 1 \leq \frac{2}{\min_i w_i}$ and that $\eta_t \leq \min_i w_i$. Therefore, we have that

$$(w_i^t)^{\alpha-1} \leq (w_i^{t-1})^{\alpha-1} + 2\zeta \leq \cdots \leq m^{1-\alpha} + 2\zeta(t-1) \leq m^{1-\alpha} + 2\zeta T.$$

Thus, $w_i^t \geq (m^{1-\alpha} + 2\zeta T)^{1/(\alpha-1)} \geq \Omega((\zeta T)^{1/(\alpha-1)})$ for all $i, t$.

We now use the standard regret bound for online mirror descent, see e.g. (Orabona, 2021, Theorem 6.8):

$$\text{Regret} \leq \max_t \frac{B_R(\mathbf{u}; \mathbf{w}^t)}{\eta_T} + \frac{1}{2\lambda} \sum_{t=1}^T \eta_t \left\|\nabla L^t(\mathbf{w}^t)\right\|_*^2 \tag{9}$$

where $B_R(\cdot; \cdot)$ is the Bregman divergence of with respect to $R$, $\mathbf{u}$ is the optimal (overall loss-minimizing) point, $\lambda$ is a constant such that $R$ is $\lambda$-strongly convex with respect to a norm of our choice over $\Delta^m$, and $\|\cdot\|_*$ is the dual norm of the aforementioned norm.

Note that for any $\mathbf{x} \in \Delta^m$, we have

$$\max_{\mathbf{v} \in \Delta^m} B_R(\mathbf{v}; \mathbf{x}) = \max_{\mathbf{v} \in \Delta^m} R(\mathbf{v}) - R(\mathbf{x}) - \nabla R(\mathbf{x}) \cdot (\mathbf{v} - \mathbf{x}) \leq \frac{m^{1-\alpha}}{\alpha} + (\min_i x_i)^{\alpha-1}.$$

In the last step, we use the fact that $-\nabla R(\mathbf{x}) = (x_1^{\alpha-1}, \ldots, x_m^{\alpha-1})$ (all of these coordinates are positive), so $-\nabla R(\mathbf{x}) \cdot (\mathbf{v} - \mathbf{x}) \leq (x_1^{\alpha-1}, \ldots, x_m^{\alpha-1}) \cdot \mathbf{v}$, and that all coordinates of $\mathbf{v}$ are non-negative and add to 1.

Therefore, given our bound on $w_i^t$, we have that this first component of our regret bound (9) is at most

$$\frac{1}{\eta_T}\left(\frac{m^{1-\alpha}}{\alpha} + m^{1-\alpha} + 2\zeta T\right) \leq O\left(\frac{\zeta T}{\eta_T}\right) \leq O\left(\frac{\zeta T}{(\zeta T)^{1/(\alpha-1)}}\right) = O\left((\zeta T)^{(2-\alpha)/(1-\alpha)}\right).$$

To bound the second term, we choose to work with the $\ell_1$ norm. To show that $R$ is $\lambda$-convex it suffices to show that for all $\mathbf{x}, \mathbf{y} \in \Delta^m$ we have $(\nabla^2 R(\mathbf{x})\mathbf{y}) \cdot \mathbf{y} \geq \lambda \|\mathbf{y}\|^2$, where $\nabla^2 R$ is the Hessian of $R$ (Shalev-Shwartz, 2007, Lemma 14) (see also (Orabona, 2021, Theorem 4.3)). Equivalently, we wish to find a $\lambda$ such that

$$(1 - \alpha) \sum_i x_i^{\alpha-2} y_i^2 \geq \lambda.$$

Since $x_i^{\alpha-2} \geq 1$ for all $i$, the left-hand side is at least $(1-\alpha) \sum_i y_i^2 \geq \frac{1-\alpha}{m}$, so $\lambda = \frac{1-\alpha}{m}$ suffices.

Now, given $\theta \in \mathbb{R}^m$, we have $\|\theta\|_* = \max_{\mathbf{x}:\|\mathbf{x}\| \leq 1} \theta \cdot \mathbf{x}$. In the case of the $\ell_1$ primal norm, the dual norm is the largest absolute component of $\theta$. Thus, we have

$$\left\|\nabla L^t(\mathbf{x}^t)\right\|_* \leq \frac{\zeta}{w_i^t} \leq O\left(\zeta(\zeta T)^{1/(1-\alpha)}\right) = O\left(\zeta^{(2-\alpha)/(1-\alpha)} T^{1/(1-\alpha)}\right).$$

Since $\eta_t \leq O(T^{-1/2})$, we have that the second component of our regret bound (9) is at most

$$O\left(T \cdot T^{-1/2} \cdot \zeta^{2(2-\alpha)/(1-\alpha)} T^{2/(1-\alpha)}\right) \leq O\left(\zeta^{2(2-\alpha)/(1-\alpha)} T^{(5-\alpha)/(1-\alpha)}\right).$$

This component dominates our bound on the regret of the first component, in both $\zeta$ and $T$. This concludes the proof. $\qquad\square$

*Proof of Corollary 4.9.* Let $Z$ be the minimum value of $\zeta$ such that $-\frac{\zeta}{w_i^t} \leq \partial_i L^t(\mathbf{w}^t) \leq \zeta$ for all $i, t$. Note that by Lemma 4.8, we have that

$$\mathbb{P}\left[Z \geq x\right] \leq \sum_{i=1}^m \sum_{t=1}^T (mn^2 e^{-\frac{x}{n}} + ne^{-x}) \leq 2m^2 n^2 T e^{-\frac{x}{n}}.$$

Let $\mu$ be the constant hidden in the big-O of Lemma A.6, i.e. a constant (dependent on $m$, $n$, and $\alpha$) such that

$$\text{Regret} \leq \mu Z^{2(2-\alpha)/(1-\alpha)} T^{(5-\alpha)/(1-\alpha)}.$$

Let $r(Z, T)$ be the expression on the right-hand side. The small gradient assumption not holding is equivalent to $Z > 12n \ln T$, or equivalently, $r(Z, T) > r(12n \ln T, T)$. The expected regret of our algorithm conditional on the small gradient assumption *not* holding, times the probability of this event, is therefore at most the expected value of $r(Z, T)$ conditional on the value being greater than $r(12n \ln T, T)$, times this probability. This is equal to

$$r(12n \ln T, T) \cdot \mathbb{P}\left[Z > 12n \ln T\right] + \int_{x=r(12n \ln T, T)}^{\infty} \mathbb{P}\left[r(Z, T) \geq x\right] dx$$

$$\leq \sum_{k=11}^{\infty} r((k+1)n \ln T, T) \cdot \mathbb{P}\left[Z \geq kn \ln T\right]$$

$$\leq \sum_{k=11}^{\infty} \mu \cdot ((k+1)n \ln T)^{2(2-\alpha)/(1-\alpha)} T^{(5-\alpha)/(1-\alpha)} \cdot 2m^2 n^2 T \cdot T^{-k}$$

$$\leq \sum_{k=11}^{\infty} \tilde{O}(T^{1+(5-\alpha)/(1-\alpha)-k}) = \tilde{O}(T^{(5-\alpha)/(1-\alpha)-10}),$$

as desired. (The first inequality follows by matching the first term with the $k = 11$ summand and upper-bounding the integral with subsequent summands, noting that $r((k+1)n \ln T, T) \geq 1$.) $\quad\square$

Note that $\frac{5-\alpha}{1-\alpha} - 10 \leq \frac{5-1/2}{1-1/2} - 10 = -1$. Therefore, the contribution to expected regret from the case that the small gradient assumption does not hold is $\tilde{O}(T^{-1})$, which is negligible. Together with Corollary 4.7 (which bounds regret under the small gradient assumption), this proves Theorem 3.2.

We now extend Theorem 3.2 by showing that the theorem holds even if experts are only *approximately* calibrated.

**Definition A.7.** For $\tau \geq 1$, we say that expert $i$ is $\tau$-*calibrated* if for all $\mathbf{p} \in \Delta^n$ and $j \in [n]$, we have that $\mathbb{P}\left[J = j \mid \mathbf{p}^j = \mathbf{p}\right] \leq \tau p_j$. We say that $\mathbb{P}$ satisfies the $\tau$-*approximate calibration property* if every expert is $\tau$-calibrated.

**Corollary A.8.** *For any $\tau$, Theorem 3.2 holds even if the calibration property is replaced with the $\tau$-approximate calibration property.*

(Note that the $\tau$ is subsumed by the big-$O$ notation in Theorem 3.2; Corollary A.8 does not allow experts to be arbitrarily miscalibrated.)

Technically, Corollary A.8 is a corollary of the *proof* of Theorem 3.2, rather than a corollary of the theorem itself.[10]

*Proof.* We only used the calibration property in the proofs of Equations (2) and (3). In the proof of Equation (2), we used the fact that $\mathbb{P}\left[J = j \mid p_j^i \le e^{-\zeta}\right] \le e^{-\zeta}$; the right-hand side now becomes $\tau e^{-\zeta}$, and so the right-hand side of Equation (2) changes to $\tau n e^{-\zeta}$. Similarly, in the proof of Equation (3), we use the calibration property in the proof of Lemma A.4; the right-hand side of the lemma changes to $\tau mnq$, and correspondingly Equation (3) changes to $\tau mn^2 e^{-\zeta/n}$.

Lemma 4.8 is only used in the proof of Corollary 4.9, where $2m^2n^2T$ is replaced by $2\tau m^2 n^2 T$. Since $\tau$ is a constant, Corollary 4.9 holds verbatim. $\qquad\square$

# B $\quad \Omega(\sqrt{T})$ **Lower bound**

We show that no OMD algorithm with a constant step size[11] substantially outperforms Algorithm 1.

**Theorem B.1.** *For every strictly convex function $R : \Delta^m \to \mathbb{R}$ that is continuously twice differentiable at its minimum, and $\eta \ge 0$, online mirror descent with regularizer $R$ and constant step size $\eta$ incurs $\Omega(\sqrt{T})$ expected regret.*

*Proof.* Our examples will have $m = n = 2$. The space of weights is one-dimensional; let us call $w$ the weight of the first expert. We may treat $R$ as a (convex) function of $w$, and similarly for the losses at each time step. We assume that $R'(0.5) = 0$; this allows us to assume that $w_1 = 0.5$ and does not affect the proof idea.

It is straightforward to check that if Experts 1 and 2 assign probabilities $p$ and $\frac{1}{2}$, respectively, to the correct outcome, then

$$L'(w) = \frac{(1-p)^w}{p^w + (1-p)^w} \ln \frac{1-p}{p}.$$

If roles are reversed (they say $\frac{1}{2}$ and $p$ respectively) then

$$L'(w) = -\frac{(1-p)^{1-w}}{p^{1-w} + (1-p)^{1-w}} \ln \frac{1-p}{p}.$$

We first prove the regret bound if $\eta$ is small ($\eta \le T^{-1/2}$). Consider the following setting: Expert 1 always reports $(50\%, 50\%)$; Expert 2 always reports $(90\%, 10\%)$; and Outcome 1 happens with probability $90\%$ at each time step. It is a matter of simple computation that:

- $L'(w) \le 2$ no matter the outcome or the value of $w$.

- If $w \ge 0.4$, then $p_1^*(w) \le 0.8$.

The first point implies that $R'(w_t) \ge -2\eta t$ for all $t$. It follows from the second point that the algorithm will output weights that will result in an aggregate probability of less than $80\%$ for values of $t$ such that $-2\eta t \ge R'(0.4)$, i.e. for $t \le \frac{-R'(0.4)}{2\eta}$. Each of these time steps accumulates constant regret compared to the optimal weight vector in hindsight (which with high probability will be near 1). Therefore, the expected total regret accumulated during these time steps is $\Omega(1/\eta) = \Omega(\sqrt{T})$.

---

[10]Fun fact: the technical term for a corollary to a proof is a *porism*.

[11]While Algorithm 1 does not always have a constant step size, it does so with high probability. The examples that prove Theorem B.1 cause $\Omega(\sqrt{T})$ regret in the typical case, rather than causing unusually large regret in an atypical case. This makes our comparison of Algorithm 1 to this class fair.

Now we consider the case in which $\eta$ is large ($\eta \geq \sqrt{T}$). In this case our example is the same as before, except we change which expert is "ignorant" (reports $(50\%, 50\%)$) and which is "informed" (reports $(90\%, 10\%)$). Specifically the informed expert will be the one with a lower weight (breaking ties arbitrarily).

We will show that our algorithm incurs $\Omega(\eta)$ regret compared to always choosing weight $0.5$. Suppose without loss of generality that at a given time step $t$, Expert 1 is informed (so $w^t \leq 0.5$). Observe that

$$L(w^t) - L(0.5) = -(0.5 - w^t)L'(0.5) + O((0.5 - w)^2)$$

$$= -(0.5 - w^t)\frac{\sqrt{1-p}}{\sqrt{p} + \sqrt{1-p}} \ln \frac{1-p}{p} + O((0.5 - w)^2),$$

where $p$ is the probability that Expert 1 assigns to the event that happens (so $p = 0.9$ with probability $0.9$ and $p = 0.1$ with probability $0.1$). This expression is (up to lower order terms) equal to $c(0.5 - w^t)$ if $p = 0.9$ and $-3c(0.5 - w^t)$ if $p = 0.1$, where $c \approx 0.55$. This means that an expected regret (relative to $w = 0.5$) of $0.6c(0.5 - w^t)$ (up to lower order terms) is incurred.

Let $D$ be such that $R''(w) \leq D$ for all $w$ such that $|w - 0.5| \leq \frac{\sqrt{T}}{4D}$. (Such a $D$ exists because $R$ is continuously twice differentiable at $0.5$.) If $|w^t - 0.5| \geq \frac{\sqrt{T}}{4D}$, we just showed that an expected regret (relative to $w = 0.5$) of $\Omega\left(\frac{\sqrt{T}}{4D}\right)$ is incurred. On the other hand, suppose that $|w^t - 0.5| \leq \frac{\sqrt{T}}{4D}$. We show that $|w^{t+1} - 0.5| \geq \frac{\sqrt{T}}{4D}$.

To see this, note that $|L'(w^t)| \geq 0.5$, we have that $|R'(w^{t+1}) - R'(w^t)| \geq 0.5\eta$. We also have that $D|w^{t+1} - w^t| \geq |R'(w^{t+1}) - R'(w^t)|$, so $D|w^{t+1} - w^t| \geq 0.5\eta$. Therefore, $|w^{t+1} - w^t| \geq \frac{\eta}{2D} \geq \frac{\sqrt{T}}{2D}$, which means that $|w^{t+1} - 0.5| \geq \frac{\sqrt{T}}{4D}$.

This means that an expected regret (relative to $w = 0.5$) of $\Omega\left(\frac{\sqrt{T}}{4D}\right)$ is incurred on at least half of time steps. Since $D$ is a constant, it follows that a total regret of at least $\Omega(\sqrt{T})$ is incurred, as desired. $\qquad\square$

