# OpenReview forum: "No-Regret Learning with Unbounded Losses: The Case of Logarithmic Pooling"
_NeurIPS.cc/2023/Conference — NeurIPS 2023 poster_

### Official Review · Reviewer_XNSv · 2023-07-03

**Soundness:** 4 excellent
**Presentation:** 4 excellent
**Contribution:** 3 good
**Rating:** 6
**Confidence:** 3

**Summary:**

This paper studies the logarithmic pooling method for prediction using expert advice. At each step, $m$ experts report distributions $p^1, p^2, \ldots, p^m$ over a size-$n$ domain. The goal is to make predictions with a vanishing regret in terms of the log loss.

The usual logarithmic pooling returns an aggregated distribution $p^*$ such that each $p^*(j)$ is proportional to $\prod_{i=1}^m[p^i(j)]^{1/m}$, the geometric mean of the predicted probabilities. This paper considers a generalization in which the exponent for each expert is replaced by $w_1, w_2, \ldots, w_m$ that sum up to 1 (instead of $1/m$ identically), and the goal is to learn the right set of weights to achieve a low regret.

Formally, each round of the prediction proceeds as follows: First, algorithm selects weights. Then, adversary chooses the experts' forecasts and the outcome subject to a *calibration* condition. The two steps are in sequential order, so that the forecaster is prevented from outputting an arbitrary distribution.

The main result of the paper (Theorem 3.2) states that the regret can be upper bounded by roughly $m^{3/2}n\sqrt{T}\log T$ in the $T \gg m, n$ regime. The algorithm uses online mirror descent with the Tsallis entropy regularizer.

**Strengths:**

The problem setting is quite natural and well-motivated. The authors did a good job in explaining why certain assumptions are necessary by providing illustrative examples. While the solution is based on online mirror descent, the analysis of the algorithm requires several new ideas that seem to be non-trivial and of independent interest.

**Weaknesses:**

- The scope of the setting is limited to logarithmic pooling and the log loss.
- The tightness of the bound is unclear; an $\Omega(\sqrt{T})$ lower bound is proved for a restricted class of algorithms, while the polynomial dependence on $m$ and $n$ might be sub-optimal.
- Several aspects of the setting don't seem sufficiently convincing; see questions below.

**Questions:**

- The setting assumes that the adversary chooses the forecasts and outcome after seeing the weights selected by the algorithm. If both parties act simultaneously, would the setting become significantly easier? (In particular, the obstacle of Example 1.1 still seems to be there?)
- What if the weights don't need to sum up to 1? Would that change the expressivity of the aggregation? (Multiplying $w_1, \ldots, w_m$ by the same factor seems to give a different distribution.)
- Lines 63--70 argue that we need to keep the algorithm from seeing the experts' advice, so that the algorithm cannot "output an essentially arbitrary function of the forecasts". What would be the practical motivation/justification of this choice?

**Limitations:**

The main limitations are the assumptions made regarding the setting (e.g., the adversarial choices of outcomes and forecasts must satisfy the calibration condition). These have been clearly pointed out in the abstract and introduction and, the assumptions are formally stated in the paper.

---

> ### Author Rebuttal · Authors · 2023-08-09
>
> Thank you for these comments. In response to the questions:
> - Question 1: Our interpretation of this question is: what would happen if we changed our setting so that the aggregation algorithm chooses weights at the same time as the adversary chooses the probability distribution, such that neither has knowledge of the other’s choice? In that case, we would seemingly end up with a game in which optimal play may involve randomized strategies. You are correct that without the calibration property, the obstacle of Example 1.1 remains. (The adversary chooses which expert reports the extreme report at random, and then the bad outcome of the example happens with probability 50%.) It would be interesting to analyze this game in the presence of the calibration property, though somewhat less standard. The setting would be less adversarial than ours, and we are able to exhibit a no-regret algorithm even in our (more adversarial) setting.
> - Question 2: Yes -- by allowing weights that do not sum to 1, new aggregate forecasts may be obtained.  Logarithmic pooling with weights adding to a constant other than 1 is attested in the literature; see e.g. [Satopaa et al., 2013] (“Combining multiple probability predictions using a simple logit model”). We conjecture that our techniques can be extended to find weights for logarithmic pooling in this broader setting, though doing so appears to be nontrivial.
> - Question 3: See our response to (3.) in the global rebuttal. Briefly, we argue that our goal is to study logarithmic pooling, and by allowing the aggregator to pick weights after seeing forecasts (i.e. allowing the weights to depend on the forecasts), the aggregation method becomes arbitrary, rather than being logarithmic pooling. Our setting is analogous to the well-studied setting of choosing weights for the optimal linear pool (see e.g. [Cesa-Bianchi and Lugosi, 2006, Section 3.3]), but for logarithmic pooling instead of linear pooling.

---

> > ### Comment · Reviewer_XNSv · 2023-08-18
> > **Thank you for your reply!**
> >
> > I would like to thank the authors for the clarification. I don't have further questions, and my overall evaluation remains positive.

---

### Official Review · Reviewer_n3Hc · 2023-07-05

**Soundness:** 4 excellent
**Presentation:** 3 good
**Contribution:** 3 good
**Rating:** 6
**Confidence:** 3

**Summary:**

Summary of the paper
====================
* The prediction setting explored in this work is harder than the usual "prediction with experts advice" (henceforth abbreviated PwE) setting (e.g. as in the Cesa-Bianchi and Lugosi book), since the learner is required to reveal the expert weights (w_t) *before* observing the expert advice (adversarially chosen by the environment) and the outcome (in the standard PwE model, the mixture prediction is to be done before observing the outcome but after observing the advice). This is crucial for the lower-bound in Example 1.1.
* The form of weighted logarithmic pooling considered in this paper to aggregate the advice of the experts (Lines 47-48) is also natural and considered in the previous literature (as minimizing weighted average KL divergence to the expert advice, having external Bayesianality, maintaining log-concavity of densities etc).
* The algorithm proposed is again efficient and well-studied in the literature; online mirror descent with the Tsallis entropy regularizer (as in Zimmert and Seldin, 2018).
* To beat the lower-bound (Example 1.1) and achieve sublinear regret, the condition imposed on the adversarial joint-sampling of expert advices and outcome is that the all the experts must be calibrated (for each expert, the conditional probability of the outcome conditioned on that expert's advice must match the advice distribution).
* With the calibration assumption and an appropriate step-size schedule, OMD+Tsallis is shown to achieve $\tilde{O}(\sqrt{T} \log T)$ regret (for $T \gg m$).

**Strengths:**

* The adversarial prediction setting explored in this work is harder than the usual PwE setting. The form of weighted logarithmic pooling considered is well-studied in the literature (especially in the context of combining Bayesian priors) and this work seems to give the first non-trivial online adversarial prediction regret bound for the logarithmic pooling with respect to the the log loss (to the best of my knowledge).
* The algorithm proposed (OMD with Tsallis regularizer) is efficient and well-studied in the literature.
* The calibration condition proposed for the experts is extensively studied in the Bayesian inference literature (in the sense that Bayesian posteriors are naturally calibrated).
* Quite a bit of detail has been provided for the proofs in the main paper, and these are reasonably sound and well-written.

**Weaknesses:**

* Not much motivation is given for the harder adversarial setting (requiring the aggregator to predict the weights before seeing the expert forecasts).
* While the calibration condition is well-motivated in the existing literature w.r.t Bayesian posteriors, the authors have not provided sufficient motivation for why it is practical when treating experts as learners (in a prediction-with-expert-advice setting). It may be particularly problematic when the experts are learners/prediction models which are decoupled from the generation of the adversarial outcomes. With the log loss, no-regret online predictors are of course possible even with adversarial outcomes (using the Laplace or Krichevsky-Trofimov estimators for instance, as in the Cesa-Bianchi and Lugosi book Chapter 9), but it may not be very useful to aggregate such no-regret experts with logarithmic pooling.
* There are no experimental results in the paper, nor are any practical applications discussed.

**Questions:**

* Lines 66-70 show how forcing the aggregator to predict before seeing the expert forecasts disallows some form of manipulation by the aggregator (mimicking a linear mixture). Could you please elaborate on why such a situation should be disallowed (when using the aggregator for either theoretical or practical purposes)?
* Could you please provide some discussion on why calibration is still practical in the PwE setting when the experts are learners which are decoupled from the adversarial generation of outcomes (i.e. the expert $i$ has fix the forecast distribution $p^i_t$ at time $t$ before seeing the adversarial outcome $j_t$)?

**Limitations:**

Not applicable.

---

> ### Author Rebuttal · Authors · 2023-08-09
>
> Thank you for these comments. A few responses:
> - Regarding the first weakness (and first question), see our response to “Question 3” in the global rebuttal. Briefly, we argue that our goal is to study logarithmic pooling, and by allowing the aggregator to pick weights after seeing forecasts (i.e. allowing the weights to depend on the forecasts), the aggregation method becomes arbitrary, rather than being logarithmic pooling. Our setting is analogous to the well-studied setting of choosing weights for the optimal linear pool (see e.g. [Cesa-Bianchi and Lugosi, 2006, Section 3.3]), but for logarithmic pooling instead of linear pooling.
> - We further justify our modeling choices (the calibration property and logarithmic pooling) in the global rebuttal. Briefly, we argue that the calibration property often holds in practice (e.g. in modern deep neural networks) and so studying prediction with advice from calibrated experts from a theoretical standpoint is well-motivated. We then argue that logarithmic pooling makes particular sense when experts are calibrated, because logarithmic pooling “takes more seriously” confident forecasts from experts, as compared with linear pooling.
> - In response to the second question: in addition to theoretical reasons to expect experts to be calibrated (see e.g. [Foster and Vohra, 1997] or [Blasiok et al., 2023] (“When Does Optimizing a Proper Loss Yield Calibration?”)), in practice we see calibration in a variety of settings. For example, modern deep neural networks are usually calibrated (see the discussion in the global rebuttal). This raises the question: how can we adapt the standard theoretical model of online prediction with expert advice -- which by default gives full power to the adversary -- to model calibrated experts? Our model attempts to do this by still giving the adversary a lot of power while constraining it just enough to guarantee that experts are calibrated.

---

> > ### Comment · Reviewer_n3Hc · 2023-08-21
> > **Response**
> >
> > Thank you for your detailed rebuttal. I would keep my score (tending towards acceptance) for now. But I will definitely consider your clarifications (which do address some of my concerns) if required during further discussions.

---

### Official Review · Reviewer_RmFD · 2023-07-07

**Soundness:** 3 good
**Presentation:** 3 good
**Contribution:** 3 good
**Rating:** 5
**Confidence:** 3

**Summary:**

This paper investigates the logarithmic pooling method for minimizing log loss and introduces the OMD algorithm utilizing Tsallis entropy as a regularizer to update weights for the logarithmic pooling method. By assuming calibrated forecasts, the paper demonstrates that the proposed algorithm ensures a sub-linear regret. Additionally, this paper establishes a corresponding lower bound, indicating that their regret bound is optimal in terms of its dependence on the number of time steps $T$.

**Strengths:**

1. Learning weights for logarithmic pooling studied in this paper is novel as far as I know, which appears to be a crucial direction, particularly due to its natural alignment with log loss.
2. The results and methodologies presented in this paper seem to be novel and the techniques used in this paper may be of independent interest to the community.
3. This paper is well-written. The algorithm is well-motivated, and the theorems presented are rigorous and sound.

**Weaknesses:**

1. It is important to include a discussion of the relevant literature on online portfolio selection in the related work, as it addresses a similar problem to this paper, namely learning parameters using log loss.
2. It is worth noting that the lower bound only applies to OMD with a constant step size, which may limit its effectiveness. Additionally, the optimality regarding the number of experts (m) and outcomes (n) should be discussed in this paper.
3. Providing a more intuitive explanation for the choice of Tsallis entropy as the regularizer would enhance the paper's clarity and value.
4. This paper can be improved by presenting a detailed example to illustrate the significance of the logarithmic pooling method with log loss.

**Questions:**

1. How about the efficiency of the algorithm? Can the algorithm be executed efficiently?
2. Why there is no need to project our decision, as stated in footnote 5?

---

> ### Author Rebuttal · Authors · 2023-08-09
>
> Thank you for these comments. We address the comments in the “Weaknesses” and “Questions” sections.
>
> Weaknesses:
> 1. Thanks for the suggestion -- we agree, and will take care to do so in the final version, should our paper be accepted. We already cite [Cover, 1991] (“Universal Portfolios”), but that’s just a start. Are there particular papers that you would recommend citing?
> 2. We agree that this is a limitation of our work, though we would like to highlight Footnote 3 from our supplement: “While Algorithm 1 does not always have a constant step size, it does so with high probability. The examples that prove [our lower bound] cause $\Omega(\sqrt{T})$ regret in the *typical* case, rather than causing unusually large regret in an atypical case. This makes our comparison of Algorithm 1 to this class [of constant step size OMD algorithms] fair.”
> 3. Thanks -- we agree. Here’s a brief intuition for our choice of Tsallis entropy. The natural first online learning algorithm to try for our problem is OMD with the negative entropy regularizer; this is equivalent to Hedge, which is a variant of multiplicative weights. Unfortunately, when attempting to use the negative entropy regularizer, we could not rule out a failure mode in which two experts alternate in making big mistakes (assigning very low probabilities to the eventual outcome) in a way that causes nearly all weight to be assigned to the mistaken expert. For example, Expert 1 makes a big mistake that causes 99% of the weight to be on Expert 2 in the next round. Then, Expert 2 makes a big mistake that causes 99.99999% of the weight to be on Expert 1 in the next round. Then Expert 1 makes a big mistake that causes 99.99999999999999999% of the weight to be on Expert 2 in the next round, and so on. This failure mode is more plausible with the negative entropy regularizer than with Tsallis entropy, because Tsallis entropy is “steeper” near the boundary of the simplex, which makes it difficult for OMD to end up assigning nearly all weight to a single expert. However, we do not have a formal proof that the negative entropy regularizer cannot be made to work.
> 4. Thanks -- we would be happy to add an example. Here’s a simple example: consider two experts and two outcomes. Expert 1 reports the distribution (98%, 2%); Expert 2 reports (50%, 50%). With equal weights, the logarithmic pool of these forecasts is (87.5%, 12.5%). The log loss is then 0.134 if Outcome 1 happens and 2.079 if Outcome 2 happens.
>
> Questions:
> 1. Yes, the algorithm is efficient, as we argue below. We can include this argument in the final version.
>
> The only nontrivial step is finding the weight vector satisfying the equation on the last line of the algorithm. To do so, it is first necessary to compute the gradient of the loss. See Equation (5) in the supplement for a formula for the gradient of the loss, which makes it clear that the gradient can be computed in time $O(mn)$. After that, we need to find the weight vector $\mathbf{w}$ that satisfies the equation on the last line.
>
> This can be done efficiently through local search: essentially, the goal is to find weights $(w_1, \dots, w_m)$ such that the vector $(w_1^{\alpha - 1}, \dots, w_m^{\alpha - 1})$ is equal to a target vector $\mathbf{v}$ plus a constant $c$ times the all-ones vector. That is, we need to simultaneously solve the equation $w_i^{\alpha - 1} = v_i + c$ for all $i$, with weights that add up to 1. (Here, the $v_i$ are knowns and the $w_i$ and $c$ are unknowns.)
>
> We start by finding $c$, by solving the equation $\sum_i (v_i + c)^{1/(\alpha - 1)} = 1$. Such a $c$ exists because  the left-hand side of this equation is continuous and monotone decreasing and goes from infinity to zero as $c$ ranges from $-\min_i v_i$ to infinity. We can solve for $c$ very efficiently, e.g. with Newton’s method. Once we know $c$, we know each $w_i$: we have $w_i = (v_i + c)^{1/(\alpha - 1)}$. This algorithm thus takes $O(mn)$ time.
>
> 2. In general, projecting the weight vector is necessary when there is no weight vector such that the gradient of R at that weight vector is equal to the target vector. However, every vector is attested as the gradient of R somewhere. We prove this in our answer to your previous question by exhibiting an algorithm for finding the weight vector.

---

> > ### Comment · Reviewer_RmFD · 2023-08-19
> > **Reply to authors**
> >
> > I would like to thank the authors for the detailed response. I would keep my score as a (weak) accept.

---

### Official Review · Reviewer_2zyV · 2023-07-09

**Soundness:** 3 good
**Presentation:** 3 good
**Contribution:** 3 good
**Rating:** 7
**Confidence:** 4

**Summary:**

The paper studies no-regret learning in the setting of logarithmic pooling of experts with the logarithmic loss. In this setting, there is a set of experts each outputting a distribution $p_i^{t}$ over outcomes in some finite set $Y$. The task of the learner is to output a vector $w^{t}$. An outcome $y$ in $Y$ is then observed and the learner suffers loss $\sum_i w^{t}_i  \log p^{t}_i (y)$ where the sum is over the experts. The loss is then compared to the best choice of weights $w$ in hindsight. Under an additional assumption that the expert probabilities are "calibrated" (which the paper argues is necessary), the paper presents $\sqrt{T}$ regret algorithm. Further, the paper is the first to consider regret in the logarithmic pooling setting and to connect calibration of experts to regret in this setting.

**Strengths:**

The main contributions of the paper in my opinion is the initiation of study of logarithmic pooling + logarithmic loss to the study of online learning. The paper justifies this setting by noting various nice properties this method of aggregation is known to satisfy. Further the relationship of calibration to the regret in this setting is also interesting. It is the key condition under which their algorithm has non-trivial regret.

**Weaknesses:**

The main issue I have with the setting is that beyond the presentation in terms of pooling, this setting seems like an instance of standard online linear optimization. As the paper notes the main difference here is the apriori unbounded sizes of the loss vectors. Viewed in this light calibration is the same as assuming that the losses are bounded ( ). The issue really is whether this is a priori clear or not. The point that makes me lean towards the fact that the calibration is "just assuming" boundedness is that no average notion of calibration (for example calibration on average across time steps) seems to be sufficient (it seems that the counterexample can be modified to be calibrated on average). Thus, the interpretation just as having "good experts" is less clear and seems more like assuming bounded losses.

**Questions:**

- It would be very helpful to elaborate the usual setting of learning with the log loss for comparison since that is well established area of online learning and information theory. The reason this would be helpful is that in the usual setting the actions are unrestricted but it turns out to be (essentially) optimal to consider "linear pooling". The fact that linear pooling comes up naturally in that setting would be a good constrast and good place to compare linear and logarithmic pooling.
- The comment about linear pooling and logarithmic regret needs to be considered a bit more carefully. In the usual notion, the learner does not compete against all convex combinations of experts. In fact, there can be polynomial difference between the two cases.

**Limitations:**

Yes

---

> ### Author Rebuttal · Authors · 2023-08-09
>
> Thank you for these comments. A few points and clarifications:
>
> * We disagree that our setting is an instance of *linear* optimization. The log loss of a forecast is not a linear function of the forecast, nor of the weights that the aggregator assigns to the various experts.
> * We disagree that the calibration property places a bound on the losses of the experts (or of the aggregate forecast). That’s because an expert’s forecast could assign an arbitrarily small probability to some outcome -- and if that outcome happens, then they could realize an arbitrarily large loss. The calibration property does guarantee that the probability of this occurring is small: in particular, that the probability of loss $\ell$ is only $\exp(-\ell)$. The fact that losses in our setting are stochastic, even if bounded in expectation, complicates the analysis and makes it far from obvious that standard learning algorithms for finding the best mixture of forecasters can be successful in our setting.
> * We would conjecture that our results could be adapted to go through for experts that are calibrated on average. The natural notion of calibration-on-average would look something like: when an expert assigns a 1% chance to an outcome, when averaged across time steps it is the case that the probability of the outcome is 1%. (Perhaps on some time steps it is 2% and on others it is 0%, but on average it’s 1%.) It seems that Example 1.1 cannot be adapted to the case of calibrated-on-average experts. For example, if the adversary chooses for an expert to assign $\exp(-T)$ probability to some outcome, then the probability of that outcome occurring is at most $T \exp(-T)$, since there are T time steps over which the expert must be calibrated on average. This is up from $\exp(-T)$ in the case of calibration at each time step, but still very small.
>
> Here are our answers to the two questions:
>
> * Thanks for prompting us to further justify logarithmic pooling. We do so in the global rebuttal (Question 2). To summarize, we believe that logarithmic pooling contrasts favorably to linear pooling in the context of calibrated experts. This is because logarithmic pooling “takes more seriously” confident forecasts from experts. Imagine two calibrated experts and three outcomes. Suppose that Expert 1 has strong evidence against Outcome 1, and so reports (0.04%, 49.98%, 49.98%), and that Expert 2 has strong evidence against Outcome 2, and so reports (49.98%, 0.04%, 49.98%). Accounting for both experts’ evidence would mean placing low probabilities on the first two outcomes and most probability mass on the third outcome. Logarithmic pooling successfully does this, returning (2.7%, 2.7%, 94.6%). By contrast, linear pooling would return (25.01%, 25.01%, 49.98%).
> * Thanks for pointing this out. On lines 152-153, we are referring to the subsection titled “A Mixture Forecaster for Exp-concave Losses” of [Cesa-Bianchi and Lugosi, 2006], specifically Theorem 3.3, which addresses the setting of competing with the best (linear) mixture of forecasters. That section addresses bounded, exp-concave loss functions. The log loss is not bounded, and we cited [Cover, 1991] for the log loss. Upon a closer look, it seems that [Cover, 1991] (cited on line 154) attains logarithmic regret in a slightly different setting (closer to the standard online convex optimization setting that of learning a mixture of experts). Do you know of a source that proves a no-regret guarantee in the prediction with expert advice setting, when choosing weights for a linear pool to compete with the best linear pool in hindsight? If not, and if we cannot find such a source, then we will take care to clarify the difference between our setting and that of [Cover, 1991].

---

> > ### Comment · Reviewer_2zyV · 2023-08-18
> >
> > We thank the author for the detailed response and apologize for the delay in the response.
> >
> > 1. Regarding the linear pooling, I do not know of an explicit reference but people do study complete against "general" experts and the comment I made was about even simple cases where "taking the convex hull" increases the regret. see https://arxiv.org/pdf/2303.07279 for an example.
> >
> > 2. I agree with the point about approximate calibration.
> >
> > 3. The reason I was saying that this was similar to linear optimization is the following: the actions are the weights $w$ and the loss suffered is $ \sum_i w_i   log(p^i_j)$ right? I understand that the loss has some semantics related to predictions but in the end is the game not similar to predicting $w$ and suffering against a loss vector $ (.. log(p^i_j) ... )$ ? This is what I was saying about in this notation calibration seems like a bound on the "size" of the loss. Am I misunderstanding something?

---

> > > ### Author Response · Authors · 2023-08-18
> > >
> > > Response to 1: Thanks for the reference! We will take a look.
> > >
> > > Response to 3: Ah yes, I believe there's a misunderstanding, and I have a guess about where the misunderstanding is coming from. The issue is that the normalizing constant $c$ in the formula for the logarithmic pool depends on the weight vector. Recall (line 30) that $c$ is the normalizing constant that makes the probabilities add to $1$. Since the logarithmic pool $p_j^*(\mathbf{w})$ is defined as $c \prod_{i = 1}^m (p_j^i)^{w_i}$, the normalizing constant $c$ can be written as
> > > $$c = \frac{1}{\sum_{j = 1}^n \prod_{i = 1}^m (p_j^i)^{w_i}},$$
> > > which depends on $\mathbf{w}$.
> > >
> > > The loss suffered by the aggregator is
> > > $$\log p_j^*(\mathbf{w}) = \log \left( c \prod_i (p_j^i)^{w_i} \right) = \log c + \sum_i w_i \log(p_j^i).$$
> > > If $c$ were independent of $\mathbf{w}$, then it would be correct to say that our problem is an instance of linear optimization. However, since $c$ depends on $\mathbf{w}$ (according to the formula above), this is not the case.
> > >
> > > I think the paper would be made clearer from us clarifying this point, and we will do so for the final version, should the paper be accepted. Thanks!

---

> > > > ### Comment · Reviewer_2zyV · 2023-08-19
> > > >
> > > > Thanks much for the clarification. This helps a lot. I am convinced that the paper is interesting. But the sake of the reader I would recommend the following changes:
> > > > - Add to the paper justification for logarithmic pooling
> > > > - Similarly for calibration. The present presentation makes it appear like logarithmic pooling is a bit ad hoc and calibration is a technical fix that get around an issue. My current understanding is that this is not the case
> > > > - A more thorough review and exposition of the connection and motivation to learning with the log loss (linear pooling as you call it). The reason this would be helpful is that the linear pooling notion is very well motivated in terms of compression and density estimation. Comparisons would hopefully explain settings where one would be better than the other and maybe also hint at whether there is an interesting deeper theory for the case of logarithmic pooling as well.
> > > > - Explicitly mention the loss and the challenges that arise due to the normalization term. For example, if possible explicitly relating the challenges to the corresponding linear optimization problem.
> > > >
> > > > I trust that the authors will make the appropriate versions of the above changes and I am increasing my score.

---

> > > > > ### Author Response · Authors · 2023-08-19
> > > > >
> > > > > Thanks for the back-and-forth and the suggestions. We will indeed make these changes.

---

### Author Rebuttal · Authors · 2023-08-09

We would like to thank all reviewers for their thoughtful comments. In this global rebuttal, we will address three recurring questions:

1. Some reviewers were interested in further justification of the calibration property.
2. Some reviewers were interested in further justification of using logarithmic pooling to aggregate experts’ forecasts, especially in light of the fact that low regret is attainable with a simple weighted average (as we mention in lines 153-154).
3. Some reviewers asked for further explanation of lines 63-70, in which we argue that at each time step, the algorithm should choose the experts’ weights without seeing the forecasts.

In the paper, we note some theoretical and empirical considerations that justify logarithmic pooling (lines 27-43) and present the calibration property as a natural condition under which there is hope for low regret in the face of challenges presented by requiring logarithmic pooling. In this rebuttal, we present an additional and orthogonal justification: we first justify the calibration property and then justify logarithmic pooling as a particularly sensible aggregation method for calibrated experts. To briefly summarize:

1. In practice, modern deep neural networks are calibrated. Our work addresses the aggregation of such networks’ outputs from a theoretical standpoint by balancing the desire for worst-case guarantees (as is typical for theory) with practical observations about neural networks’ performance.
2. Logarithmic pooling makes sense *in particular* when experts are calibrated, because it takes confident predictions seriously, especially as compared with linear pooling.
3. Learning optimal weights for linear pooling is a well-studied question. Our argument in favor of logarithmic pooling suggests addressing the same question for logarithmic pooling. However, if the weights to a logarithmic pool are allowed to depend on the forecasts, then the aggregation method is no longer a logarithmic pool; it can be an arbitrary function.

Elaborating point by point:

1. In lines 85-89, we give two theoretical justifications for the calibration property. Here we note another justification: in practice, modern deep neural networks are well-calibrated when trained on a proper loss function such as log loss. This is true for a variety of tasks, including image classification and language modeling. (See e.g. [Blasiok et al., 2023] (“When Does Optimizing a Proper Loss Yield Calibration?”) or [Kadavath et al., 2022] ("Language Models (Mostly) Know What They Know").)

Now, suppose that we wish to use an ensemble of off-the-shelf neural networks for some prediction or classification task. We trust the networks to be calibrated, but we may not know beforehand which networks have the highest levels of expertise. (As an extreme example, suppose there are a thousand classes, all equally likely. A network that always assigns 100% probability to the correct class has more expertise than a network that always assigns a uniform distribution over the classes, even though both are calibrated.) Learning to aggregate these networks’ forecasts is well-motivated and nontrivial. The standard fully adversarial setting is one reasonable (and well-trodden) theoretical approach to this problem. However, integrating the empirical observation that neural networks are well-calibrated into the theoretical analysis yields a new, well-motivated setting with the potential for stronger results than are possible without assuming calibration.

2. In lines 27-43, we gave a few theoretical and empirical justifications for using logarithmic pooling. We believe that logarithmic pooling makes *particular* sense when the experts are calibrated. Intuitively, this is because the logarithmic pool pulls the aggregate toward more confident forecasts. We give one example in lines 32-33: if Expert 1 reports probability distribution (0.1%, 99.9%) over two outcomes and Expert 2 reports (50%, 50%), then the logarithmic pool (with equal weights) is roughly (3%, 97%). This makes more sense than the linear pool (i.e. arithmetic average), which would be roughly (25%, 75%): if Expert 1 is calibrated (as we have assumed and justified above), then a (0.1%, 99.9%) forecast entails very strong evidence in favor of Outcome 2 over Outcome 1. Meanwhile, Expert 2’s forecast gives no evidence either way. An aggregate of very strong evidence for Outcome 2 with no evidence either way ought to still be fairly strongly in favor of Outcome 2. (Indeed, there is a natural interpretation of logarithmic pooling in terms of averaging experts’ Bayesian *evidence*; see [Neyman and Roughgarden, 2023].)

As another example, suppose that Expert 1 reports (0.04%, 49.98%, 49.98%) and Expert 2 reports (49.98%, 0.04%, 49.98%) (a natural interpretation: Expert 1 found strong evidence against Outcome 1 and Expert 2 found strong evidence against Outcome 2). If both experts are calibrated, a sensible aggregate should assign nearly all probability to Outcome 3. Linear pooling does not accomplish this; meanwhile, logarithmic pooling returns roughly (2.7%, 2.7%, 94.6%), which is much more sensible.

3. The problem of choosing weights for a *linear* pool to compete with the best weights in hindsight is well-studied (see e.g. [Cesa-Bianchi and Lugosi, 2006, Section 3.3]). Above, we justified the logarithmic pool as a better alternative to linear pooling in the case of calibrated experts. It is then natural to study the analogous problem for logarithmic pooling in place of linear pooling; this is the problem we study. If the algorithm’s weights were allowed to depend on the experts’ forecasts, the resulting probability distribution (as a function of experts’ forecasts) might look nothing like logarithmic pooling. If the algorithm is to compete with the benchmark of a “best possible logarithmic pool”, it makes sense to restrict the algorithm to (weighted) logarithmic pools of the experts’ reports.

---

### Decision · Program_Chairs · 2023-09-21

**Decision:**

Accept (poster)

**Comment:**

This paper considers online learning with logarithmic pooling and introduces the first sqrt(T) regret bound. The authors are encouraged to take into account the promised changes (in particular clarification regarding relationship to online linear optimization).